# Last iterate convergence of SGD for Least-Squares in the Interpolation regime

**Aditya Varre**
EPFL
aditya.varre@epfl.ch

**Loucas Pillaud-Vivien**
EPFL
loucas.pillaud-vivien@epfl.ch

**Nicolas Flammarion**
EPFL
nicolas.flammarion@epfl.ch

## Abstract

Motivated by the recent successes of neural networks that have the ability to fit the data perfectly *and* generalize well, we study the noiseless model in the fundamental least-squares setup. We assume that an optimum predictor perfectly fits the inputs and outputs $\langle \theta_*, \phi(X) \rangle = Y$, where $\phi(X)$ stands for a possibly infinite dimensional non-linear feature map. To solve this problem, we consider the estimator given by the last iterate of stochastic gradient descent (SGD) with constant step-size. In this context, our contribution is twofold: (i) *from a (stochastic) optimization perspective*, we exhibit an archetypal problem where we can show explicitly the convergence of SGD final iterate for a non-strongly convex problem with constant step-size whereas usual results use some form of average and (ii) *from a statistical perspective*, we give explicit non-asymptotic convergence rates in the over-parameterized setting and leverage a *fine-grained* parameterization of the problem to exhibit polynomial rates that can be faster than $O(1/T)$. The link with reproducing kernel Hilbert spaces is established.

## 1 Introduction

As soon as large-scale statistics and optimization need to work together, stochastic gradient descent (SGD) is the core algorithm everybody tries to build upon [7]. Its versatility, practicability and adaptability make it the workhorse of almost every supervised machine learning problem. Yet, its outstanding efficiency remains mysterious, or at least surprising on certain aspects. Furthermore, the recent successes of deep neural networks (DNN) brought a new paradigm to the classical supervised learning setting with the ability to fit the data perfectly *and* to generalize well [5]. Following this idea, the old statistical modelling where the model suffers from problem-dependent noise has to be revisited: there is a need of analyzing stochastic algorithms in this new light [18]. Whether we call it *over-parameterization* to put emphasis on the large number of neurons needed in DNNs, *interpolation* as in approximation theory or *noiseless model* to stress the absence of noise in this statistical model, all these terminologies refer to the same idea. This regime brings with it new insights that reflect better the current machine learning setup.

Hence, the main question: how would SGD profit from this noiseless model? At first glance, the story seems clear: the old problem of variance at optimum making the SGD iterates oscillate asymptotically now disappears. Thus, should also disappear techniques that prevent from this, namely, averaging and decaying step sizes: one should be able to study the convergence of the SGD *final iterate* with *constant step-size*.

35th Conference on Neural Information Processing Systems (NeurIPS 2021).

However, the study of the last iterate of SGD has always caused some technical problems preventing from a clear theory in this case [26]. Indeed, the convergence of the final iterate is much more difficult to prove than the convergence of the average of the iterates [27, 13, 15]. This counter-intuitive difficulty can be explained by the fact that the interactions coming from the sampling noise of SGD prevent the loss of the final iterates from decreasing. Therefore standard Lyapounov strategies often failed in such a setting. Besides, even if averaged SGD has shown *theoretically* some good convergence properties, the final iterate is commonly used *in practice*. Finally, averaging techniques always suffer from saturation coming from the slow forgetting of initial conditions.

To tackle these questions, we consider the simplest setting of the linear regression over features $\phi(X)$ in a Hilbert space $\mathcal{H}$. In this context, the setup corresponds to the existence of linear relationship between the output and the input: $Y = \langle \theta_*, \phi(X) \rangle$. Note that this setting is rich as the features can be a non-linear transformation of the inputs, as it is commonly the case when they are defined through a positive-definite kernel $K(X, X')$ [25, 28]. Note also that our analysis will, unless stated explicitly, be conducted in a non-parametric and dimensionless fashion, enabling the features to come from a infinite dimensional reproducing kernel Hilbert space. In this perspective, the problem we are considering is not strongly convex.

**Main contributions.** The aim of the present article is to answer at once the two following problems: (i) *from a (stochastic) optimization perspective*, the goal is to exhibit an archetypal problem where we can show explicitly the convergence of SGD final iterate for a non-strongly convex problem with constant step-size and (ii) *from a statistical/machine learning perspective*, the aim is to push deeper the study of the over-parameterized setting for the non-parametric least-squares problem. Contrary to the noisy setting, where (almost) everything is known, the noiseless model suffers from a certain lack of understanding. More precisely, our contributions are the following:

- We show that the final iterate of constant step-size SGD achieves a convergence rate of $O(\ln(T)/T)$ under minimal assumptions. With a slightly stricter hypothesis we improve this convergence rate to $O(1/T)$.
- Going further, we assume the usual non-parametric *capacity and source conditions* respectively on the spectrum of the covariance and the optimum, and derive bounds for this fine-grained model. In this setup, we show the SGD *fast rates* of $O(1/T^{1+\alpha})$.
- We derive an explicit recursion on the eigenspaces of the covariance matrix that is of its own interest. Indeed, it is the cornerstone of our analysis and could be very useful in the future to understand important properties of SGD.

From a technical standpoint, going from the average to the last iterate is far from anodyne for constant step-size SGD even in the interpolation regime. Indeed, in a similar setting, D.Aldous considered it an open problem [1, Sec. 3.3, Open problem (i)]. Prior to our result, how to directly deal with the fluctuations induced by the stochasticity of the gradients without any variance reduction was not known. Our work presents a direct Lyapunov technique that handles them without any explicit variance reduction methods like averaging or step-size decay.

## 1.1 Related Work

As recalled earlier, it is often easier to show convergence for the averaged iterates of SGD than for the final one. However, there has been a huge effort by the optimization and machine learning communities to work on the final iterate. We report here the different works and contexts for which such results are shown. All the results stated below are for convergence in function value.

**Last versus averaged iterates.** First, for *non-smooth functions* (and variance of the SGD-gradients uniformly bounded), it is easy to show, with proper decrease of the step sizes, that the averaged iterates converge at rate $O(1/\sqrt{T})$ in the non-strongly convex case and $O(1/T)$ when the function is strongly convex. On the other side, convergence of the last iterate has been first addressed by [35, 27] who showed convergence rates of $O(\log(T)/\sqrt{T})$ for non-strongly convex objectives and $O(\log(T)/T)$ for strongly convex objectives. These rates have recently been shown to be tight by [13] (the $\log$ term cannot be suppressed for the last iterate). Secondly, for *smooth functions* (gradient Lipschitz and variance bounded only at optimum), [2] showed in the non-strongly convex case that smoothness does not help; results on the averaged iterate are the same $O(1/\sqrt{T})$ and results on the last iterate are actually worse: $O(T^{-1/3})$. In the strongly convex setting, the optimal rate $O(1/T)$ rate is obtained both by the averaged and last iterates. However the averaged iterates are preferred

since they lead to the optimal covariance and the step size is independent of the strong convexity constant [23]. Finally, the final iterate of SGD, despite its ubiquitous use in machine learning, never theoretically performs as well as the averaged, except when used with a geometrically-decaying learning-rate [15, 12].

**Over-parameterized setting.** As recalled earlier, SGD in the over-parameterization regime corresponds to the assumption that all the function gradients vanish at optimum. It has been first studied by [24] who assumed furthermore a strong growth condition (*SGC*) (first introduced by [29, 32]) but often too stringent to match the machine learning practical set-up. Up to our knowledge, [18] was the first to point out that this interpolation regime was particularly relevant in the recent deep learning framework and to show linear convergence in the strongly-convex setting. In the non-strongly convex case, $O(1/T)$-convergence has been shown by [33] for the averaged iterates. Consequently, there has been many papers discussing this setting, proving convergence rates in different contexts: Polyak-Lojasiewicz [4], accelerated [33, 17], second-order [19], line-search [34]. However, no rate for the final iterate has been shown in the non-strongly convex setting for the last iterate as the only convergence rates achieved, $O(1/T)$, corresponds in all these works to the averaged iterates.

**The Quadratic Case.** More specifically, the quadratic case has been widely studied as a central model in the machine learning literature. As explained above, on the one hand, in the usual noisy setting, averaging or decaying step sizes have been considered, showing well-known $O(1/T)$ convergence rates [3]. The final iterate convergence of SGD for the noisy quadratic case is studied in [12] where they show a $O(d \log T/T)$ rate and advocate the primacy of exponential step decay. On the other hand, the SGD in over-parameterized setting has been considered only lately by [18, 6, 36]. The first work corresponds to the strongly convex case, where the final iterate provably converges whereas our work is more aligned to [6, 36] where the non-strongly convex case is considered. In [21], the authors also studied the noiseless setting and provide worst and average cases asymptotic rates for the non-strongly convex case in the $n, d \to \infty$ regime. In [6], among other results, it is shown that only the minimum of the function value iterates converge at rate $O(1/T)$ and up to $O(1/T^{1+\alpha})$ for $\alpha > 0$ when usual *capacity and source* conditions are assumed for the problem [8, 30]. A part of our work can be seen as a continuation of [6], strengthening its result by proving that the final iterate converges instead of the $\min$ or some average.

## 2 Problem Set-up: Stochastic Gradient Descent on the Least-squares Problem

The setting is classical for stochastic gradient descent for linear least-squares in a Hilbert space $\mathcal{H}$. The function we would like to minimize over $\theta \in \mathcal{H}$ is

$$\mathcal{R}(\theta) = \frac{1}{2}\mathbb{E}_\rho \left( \langle \theta, \phi(X) \rangle - Y \right)^2, \tag{1}$$

when $\rho$ is the joint law over input/outputs $(X, Y) \in \mathcal{X} \times \mathbb{R}$ and $\phi$ is a feature map from $\mathcal{X}$ to $\mathcal{H}$. For the sake of simplicity, the reader can assume that $\mathcal{H}$ is a finite-dimensional Euclidean space. However, we take a special care that all the quantities and definitions could be valid whenever $\mathcal{H}$ is infinite dimensional. Hence, all the theorems are valid in a Reproducing Kernel Hilbert Spaces (RKHS) framework. Without any loss of generality, and to avoid heavy notations, we assume that $\phi(X) = X$, and as soon as we refer the the finite dimensional case, we set $\mathcal{H} = \mathbb{R}^d$.

**Covariance matrix.** We define the covariance operator on $\mathcal{H}$: $H := \mathbb{E}_\rho[X \otimes X]$. This positive semi-definite operator is diagonalizable and plays a central role in our analysis. Hence, let us denote $\lambda_i$ its non-negative eigenvalues sorted in non-increasing order and $v_i$ its corresponding eigenvectors. In finite dimension, the covariance operator is simply a $d \times d$ matrix $H = \mathbb{E}_\rho[XX^\top]$ but, once again, notice that the analysis can also afford being infinite dimensional. In either case we denote $\lambda_{\max}$ the largest eigenvalue of $H$. In the finite dimensional case $\lambda_{\min} > 0$ is always correctly defined but can be arbitrarily small. On the opposite, a standard and important consequence of the infinite dimension is that, in most cases (especially when $\text{Tr} H < +\infty$), the infinite sequence of eigenvalues is no longer lower bounded as it converges to 0: $H$ is no longer invertible and the problem is non-strongly convex.

**Noiseless model.** We assume that the model does not suffer from any noise: there exists $\theta_* \in \mathcal{H}$ such that, $\rho$-almost surely, $\langle \theta_*, x \rangle = y$. This means that the model is well specified *and* that there is no noise at optimum. Due to this noiseless condition, we expect our iterates to go to the optimum

without decaying step-sizes or averaging. We can also rewrite the Risk in Eq. (1):

$$\mathcal{R}(\theta) = \frac{1}{2}\left(\theta - \theta_*\right)^\top H\left(\theta - \theta_*\right) = \frac{1}{2}\mathrm{Tr}\left[\left(\theta - \theta_*\right)\left(\theta - \theta_*\right)^\top H\right]. \tag{2}$$

**Link between Noiseless model and Interpolation.** Interpolation corresponds to the case where our predictor can fit a finite set $n$ of *training data* i.e., $\langle\theta_*, x_i\rangle = y_i$ for $i \leq n$. Note that this setting is naturally included in our model replacing the test risk $\mathcal{R}(\theta)$ with the training loss whose finite sum structure can be rewritten $\mathbb{E}_{\hat{\rho}}\left(\langle\theta, x\rangle - Y\right)^2$ where $\hat{\rho}$ is the empirical distribution of the data. Consequently our result on the final iterate convergence holds in this model for the training loss.

**SGD with constant step-size.** To minimize the function $\mathcal{R}$ defined in Eq. (1), we do not have access to the distribution $\rho$ but to a stream of i.i.d. observations $(x_t, y_t)_{t \geq 1}$ sampled from it. We hence perform a gradient descent in the direction given by one sample at a time with *constant* stepsize $\gamma > 0$. Throughout all this paper, the initial condition is always set to $\theta_0 = 0$ and for $t \geq 1$,

$$\theta_{t+1} = \theta_t - \gamma\left(\langle\theta_t, x_t\rangle - y_t\right)x_t. \tag{3}$$

**The impossibility of linear rates.** We stress that the least-squares setup we consider is a *non-strongly* convex problem. Indeed, in finite dimension $d$, $\lambda_{\min} > 0$, and SGD converges at linear rate $\sim e^{-\gamma\lambda_{\min}T}$ [18, Theorem 1], this asymptotic regime occurring after a time scale $\tau \sim 1/(\gamma\lambda_{\min})$. However, this apparent strong convexity is a lure, since the large (or infinite) dimension makes this convergence rate vacuous for an arbitrarily small $\lambda_{\min}$. Hence we focus on the non-parametric non-strongly convex setup where we prove non-asymptotic polynomial rates. Yet in finite dimension, it is always possible to see the linear regime after time $1/(\gamma\lambda_{\min})$ as shown in Figure 1, even if this time can be arbitrarily long.

## 3 Convergence rates of the final iterate of SGD

In large dimension, two quantities govern principally the rates of convergence of least-squares estimators: (i) the spectrum of the covariance matrix, and (ii) how the solution, $\theta_*$, projects on the eigenbasis of the covariance matrix. In this section, we state refined assumptions on the spectrum of the covariance matrix and the decomposition of $\theta_*$ over its eigenbasis. Note that in finite-dimension, all these quantities are always finite, but can be extremely large compared to the sample size $T$. This is why the reader can see these more as a *fine-grained parameterization* of the problem rather than restrictive assumptions. The assumptions we make always go in pairs, (i) one for the features through the covariance matrix (Assumptions 1, 3 and 5) and (ii) one for the target solution (Assumptions 2, 4 and 6).

**Summary on the results.** As assumptions go stricter, the convergence rates go faster. Every theorem is a bound on the *expected risk* given by the *last iterate* of the SGD recursion with *constant* step size $\gamma$, started at $\theta_0 = 0$. Only for Theorem 1, for which the assumptions are the weakest, we allow the step-size to depend on the finite horizon $T$ (through its logarithm). All the theorems are stated with respect to finite constants defined thanks to the different assumptions. The reader can refer to Table 1 for a concise summary of them. Note also that we put a particular effort for the clarity of the bounds, and hence, some numerical constants might appear large. These are simple artifacts on the proofs and could be lowered, but at the price of less clear results. A more detailed summary of our results can be found in Appendix 7. All the proofs are deferred to Appendix 9.

### 3.1 Standard Least-Squares setting

As often, when we analyze SGD, we make a 4-th order assumption on the distribution of the features.

**Assumption 1 (Fourth Moment Condition)** *There exists a finite constant $R \geq 0$, such that*

$$\mathbb{E}\left[\|X\|^2 \, XX^\top\right] \preccurlyeq RH \tag{4}$$

The assumption holds in the case of bounded features, i.e. when $\|X\|^2 \leqslant R$, $\rho_X$-almost surely. It also holds more generally for features with infinite supports, such as sub-Gaussian data, and canonical basis distributions (i.e. $x = e_i$ with probability $p_i$). This is a standard assumption when analysing SGD for least squares [3, 14, 12] which is weaker than what is assumed in [10, 11, 36].

| Theorem | Assumption | Condition | Rate |
|---------|-----------|-----------|------|
| Theorem **1** | A. **1** | $\mathbb{E}\left[\left\|X\right\|^2 XX^\top\right] \preccurlyeq RH$ | $\dfrac{\ln(T)}{T}$ |
| | A. **2** | $\|\theta_*\|_{\mathcal{H}} < +\infty$ | |
| | $\Uparrow$ | $\Uparrow$ | |
| Theorem **2** | A. **3** | $\mathbb{E}\left[\left\langle X, \ln\left(H^{-1}\right)X\right\rangle XX^\top\right] \preccurlyeq R_{\ln}H$ | $\dfrac{1}{T}$ |
| | A. **4** | $\mathrm{Tr}(M_0 \ln(H^{-1})) < +\infty$ | |
| | $\Uparrow$ | $\Uparrow$ | |
| Theorem **3** | A. **5** | $\mathbb{E}\left[\left\|H^{-\alpha/2}X\right\|^2 XX^\top\right] \preccurlyeq R_\alpha H,\ \alpha \in (0,1)$ | $\dfrac{1}{T^{1+\alpha\wedge\beta}}$ |
| | A. **6** | $\mathrm{Tr}(M_0 H^{-\beta}) < +\infty,\ \beta \geqslant 0$ | |

Table 1: Table showing the main results of the article: different upper-bounds for the convergence of the SGD final iterate determined by different assumptions.

**Assumption 2 (Attainable case)** *The target solution $\theta_*$ lies in the space $\mathcal{H}$. This ensures that it has a finite norm $\|\theta_*\|_{\mathcal{H}} < +\infty$.*

While this is always true in finite dimension, this assumption draws the attention on the fact that the norm of the optimum $\theta_*$ could be very large in high-dimension. As a limit, in infinite-dimensional spaces, $\theta_*$ could not belong to $\mathcal{H}$ and hence would have an infinite norm. This is why we refer to this assumption as the *attainable case*. Under these two assumptions we have the following result.

**Theorem 1** *Assume Assumptions 1, 2. Then, for $T \geqslant 2$, if we set $\gamma = (4R\ln(T))^{-1}$, we have the following bound for the expected risk of the estimator given by the $T^{th}$ iterate of SGD:*

$$\mathbb{E}\mathcal{R}(\theta_T) \leqslant 3R\|\theta_*\|_{\mathcal{H}}^2 \frac{\ln(T)}{T}. \tag{5}$$

Let us comments upon this result proven in Appendix 9.1. The theorem above states that, under mild assumptions, if we allow the step-size to depend on the time horizon, we have a $O(\ln(T)/T)$ convergence rate for the final iterate. First, removing the dependence on finite horizon $T$ for the step-size by considering decaying step-sizes $\gamma_t \propto 1/\ln(t)$ could be done, but we decided to keep this way as we have focused on constant step-size in this paper. Furthermore, Theorem 2 of [6] shows that this bound is optimal up to $\log(T)$ for a SGD. This rate can also be compared to the classical optimization results for the non-strongly convex objective (our case). It is well known that gradient descent and averaged SGD (even for non-quadratic objectives) achieve a $O(1/T)$ rate with constant step size. Similarly [6] achieves a convergence rate of $O(1/T)$ rate with constant step size for the $\min$ of the function value along the iterates. However, the rate of convergence of SGD last iterate was an open problem [see 6, Remark 1]. Note however that our bound suffers from a $\log(T)$ both in function value and for the step-size. Whether this term is necessary is an open question for us. The purpose of the following development is to first remove these $\log(T)$ dependence at the price of a log-scale refinement of the assumptions.

### 3.2 A logarithm-scale refinement

This second sequence of assumptions is slightly stronger. They reinforce assumptions 1 and 2 at the log-scale. Once again, there is one on the features and the other one is on the target.

**Assumption 3** (log-**regularity of the features**) *There exists constant $\lambda_{\mathrm{o}} > 0$ and $R_{\ln}$ such that the covariance matrix $H$ satisfies the following condition:*

$$\mathbb{E}\left[\left\langle X, \ln\left(\lambda_{\mathrm{o}}H^{-1}\right)X\right\rangle XX^\top\right] \preccurlyeq R_{\ln}H.$$

Note that $\lambda_{\mathrm{o}}$ is a just a reference which lets us present cleaner results. We choose $\lambda_{\mathrm{o}}$ such that $7R_{\ln} \leqslant \lambda_{\mathrm{o}}$. This is always possible as $R_{\ln}$ scales linearly with $\ln \lambda_{\mathrm{o}}$ (see details in 7.1). This condition implies in fact $\mathrm{Tr}\left(H \ln(\lambda_{\mathrm{o}}H^{-1})\right) \leqslant R_{\ln}$, which consequently implies the following eigenvalue decay: $\lambda_i \leqslant O(1/(i\ln i))$ (see details in 7.1). This is the reason why we say that this assumption refines Assumption 1 at a log-scale.

**Assumption 4 (log-regularity of the optimum)** *The covariance matrix at optimum $M_0 = \theta_* \theta_*^\top$ satisfies the following condition:*

$$\mathrm{Tr}\left(M_0 \ln\left(H^{-1}\right)\right) < +\infty.$$

We define $C_{\ln} := \sum_i m_i^0 \ln(\lambda_o/\lambda_i)$, with the same reference $\lambda_o$. To give an order of magnitude in finite dimensions, when the spectrum of the covariance matrix in lower bounded by $\lambda_{\min} > 0$, $C_{\ln}$ can be estimated as $C_{\ln} \leqslant \|\theta_*\|_{\mathcal{H}}^2 \ln \lambda_o/\lambda_{\min} \sim a\|\theta_*\|_{\mathcal{H}}^2 \ln d$ even if $\lambda_{\min}$ is as small as $1/d^a$. When the dimension is not too big (even at the log-scale), this constant $C_{\ln}$ is comparable with $\|\theta_*\|_{\mathcal{H}}^2$ up to log factors. Under these two assumptions we have the following convergence result.

**Theorem 2** *Assume Assumptions 3,4. Then, for $T \geqslant 3$, if we set $\gamma = (14R_{\ln})^{-1}$, we have the following bound for the expected risk of the estimator given by the $T^{th}$ iterate of SGD:*

$$\mathbb{E}\mathcal{R}(\theta_T) \leqslant \frac{10R_{\ln}C_{\ln}}{T}. \tag{6}$$

This theorem, proven in Appendix 9.2, states that at the price of a $\log$-scale refinement on the features and optimum, the SGD-convergence rate is $O(1/T)$. This naturally restricts the class of problems that suffers from a $\log(T)$ (see Theorem 1) to a very small class: roughly speaking, it is the class of problems for which the eigenvalue decreasing rate is strictly squeezed between: $O(1/(i \ln i)) < \lambda_i < O(1/i)$. The role of the assumptions is fundamental here: Assumption 3 allows to remove the $\ln(T)$ for the step-size and Assumption 4 allows to remove the $\ln(T)$ for the convergence rate. The proof technique is the same as for the previous theorem, the only difference is that the assumptions allow to control more precisely the bias and the variance in the SGD recursion.

### 3.3 A fine-grained parameterization of the problem: *capacity and source* conditions

The final set of assumptions have been introduced by [8]. They are in the same vein as above assumptions and are often called *capacity* and *source* conditions in the reproducing kernel Hilbert spaces community.

**Assumption 5 (capacity condition: $\alpha$-regularity of the features))** *The covariance matrix $H$ is such that there exists $\alpha > 0$ and a finite constant $R_\alpha > 0$ verifying*

$$\mathbb{E}\left[\langle X, H^{-\alpha}X\rangle XX^\top\right] \preccurlyeq R_\alpha H.$$

Note that Assumption 5 is strictly more demanding than Assumption 3 and has been named *regularity of the features* in [6, Remark 3]. Note also that $\alpha \to 0^+$ corresponds to Assumption 1 and the larger the $\alpha$ the stricter the assumption is. The capacity condition stated above implies $\mathrm{Tr}\left(H^{1-\alpha}\right) \leqslant R_\alpha$. which consequently implies an eigenvalue decay as a power law for the sequence of eigenvalue of $H$: $\lambda_i = O(1/i^{\frac{1}{1-\alpha}})$ (see details in 7.1). It is also often related to the *effective dimension* of the problem [8]. Finally, as a limiting case, when $\alpha \to 1$, $R_\alpha \to \mathrm{Tr}(\mathrm{Id}) = d$ and the assumption is only valid in finite dimension. Hence, it is expected that the bound would blow in large dimension when $\alpha$ grows to one. For this reason, in the literature [8, 30], the bound is often reparameterized as $\alpha' = (1-\alpha)^{-1}$ to push this singularity at infinity. We decide to keep our parameterization for the clarity of the result. The same type of assumption can be stated for the optimum.

**Assumption 6 (source condition: $\beta$-regularity of the optimum))** *The covariance matrix at optimum $M_0 = \theta_* \theta_*^\top$ is such that there exists $\beta > -1$ such that $\mathrm{Tr}(M_0 H^{-\beta}) < +\infty$. Here, we define*

$$C_\beta := \mathrm{Tr}\left(H^{-\beta}M_0\right) = \sum_i \lambda_i^{-\beta}m_i^0.$$

Here are important remarks on this assumption. First note that

$$\mathrm{Tr}(M_0 H^{-\beta}) = \mathrm{Tr}(\theta_*\theta_*^\top H^{-\beta}) = \theta_*^\top H^{-\beta}\theta_* = \|H^{-\beta/2}\theta_*\|_{\mathcal{H}}^2.$$

Hence, $\beta = 0$ corresponds to Assumption 2 which is the attainable case. It is worth noting that for $\beta \in (-1, 0)$, this assumption takes into account the fact that $\theta_*$ does not necessarily belong to $\mathcal{H}$. In such a case, it is still possible to define properly $\theta_*$ as the infimum over $\mathcal{H}$ of the risk $R$ defined in

Eq.(1). However, for sake of clarity and to avoid cumbersome technicalities, we refer to [9, p.1395] for the extension of the analysis is this case. As before, the larger the $\beta$ the stricter the assumption is. Finally as a limiting case, $\beta \to +\infty$ corresponds to the fact that $\theta_*$ belongs to a finite dimensional space. This hypothesis is often called the *source condition* in the literature because it quantifies the complexity of the optimum [see 8, 30, 20, for further details]. Once again, parameterization has not yet been fixed by the literature and one often uses the parameter $r$ that corresponds to $r = \frac{\beta+1}{2}$. We have the following theorem under these capacity and source conditions.

**Theorem 3** *Assume Assumptions 5, 6 with constants $\alpha \in (0,1)$ and $\beta > -1$ respectively. Then, for $T \geqslant 3$, we have the following bound for the expected risk of the $T^{th}$ iterate of SGD:*

$$\mathbb{E}\mathcal{R}(\theta_T) \leqslant 2C_\beta \left( \frac{1+\beta}{\gamma} \right)^{1+\beta} \frac{1}{T^{1+\alpha\wedge\beta}}, \tag{7}$$

*where $\gamma^{1-\alpha} \leqslant (32\xi_\alpha R_\alpha)^{-1}$ and $\xi_\alpha = \sum_{n \geqslant 1} \frac{1}{n^{1+\alpha}}$.*

Let us comment this result. Its proof can be found in Appendix 9.3.

**Discussion on the limit cases for $\alpha$ and $\beta$.** Let us note a few observations on the two parameters $\alpha$ and $\beta$. Firstly, it is observed in [6] that the rate $1/T^{1+\alpha\wedge\beta}$ is optimal in terms of power law for SGD. Second, the convergence rate depends on the minimum of the two. It implies that either the regularity of the features, or the one of the optimum, is a bottleneck for the convergence of SGD. More precisely if $\alpha < \beta$, the features are not regular enough to counter the multiplicative noise of SGD. Conversely, when the optimum is the bottleneck, there is no difference between SGD and Gradient Descent (GD), as $T^{1+\beta}$ is the rate of convergence of GD. Note that $\alpha \to 0$ corresponds to the setting of Theorem 1 where a $\log(T)$ appears. Hence, it is normal that our bound does not hold in this limit: indeed, we remark that the step-size shrinks to 0 as $\xi_\alpha$ goes to infinity. Another interesting limit is the case where $\alpha \to 1$: as said before this cannot occur in infinite (or large) dimension as $R_\alpha \to$ "$d$". Therefore the bound blows up in this limit also. The fact that the step-size depends on $\alpha$ is a weakness of our result and is due to the mixing power of the covariance eigenspaces. This dependence could be eliminated with an extra assumption on a lower bound on the decrease rate of the spectrum of the covariance matrix as made by [8]. Finally note that the rate of convergence is always strictly better than $1/T$ for $\beta > 0$ and is remarkably adaptive to the possible misspecification of the problem $\beta \in (-1,0)$.

**Comparison with the literature.** We can first compare to the results on the noisy setting studied by [9]. Naturally, when some additive noise is assumed, averaging is necessary, and the results are weaker: $\gamma$ is not adaptive to the problem, depends on the time-horizon and the rates are always slower than $O(1/T)$. However, when the noise is 0, the averaged iterates achieves the exact same convergence rate as in our theorem. The closest results to our theorem are in [6], where the same rates are shown. The important difference is that their results are given for the min of the function value and not the final iterate: our theorem solves an open problem stated in the aforementioned paper. Remark however that one superiority of [6] is that the step-size does not depend on $\alpha$ showing some adaptivity with this parameter. As noted in the previous paragraph, we could fix this difference with an extra assumption on the spectrum of the covariance matrix. Finally, rates of convergence for the noiseless setting have been addressed by [16] with some truncated version of the kernel ridge estimator. It was an open problem stated by the author to understand whether SGD could achieve these non-parametric rates. As for the min, this problem seems to be properly solved now. A main difference is that the bounds of [16] suffer from some saturation in the well specified setting, when $\beta > 0$, whereas our estimator does not. When $\alpha - 1 \leqslant \beta \leqslant 0$, then the bounds match, but when the problem is really misspecified, for $\beta \leqslant \alpha - 1$, the bound of [16] is strictly better than ours. As we know that our result is optimal for SGD, it raises the interesting question whether some form of acceleration [10] or multiple passes over the data [22] could reach these rates.

**Link with kernel regression.** These capacity and source conditions are classical in the reproducing kernel Hilbert spaces community. For sake of clearness, we did not mention any specificity on the features. However, as it has been done in [31, 9], SGD can be "kernelized" and the descent becomes an optimization algorithm in the RKHS space of functions. In this context, the *capacity and source* conditions take on their full meaning: when RKHS are Sobolev spaces, they represent a smoothness condition on the features and the optimum respectively. The coefficient $\alpha$ would be the degree of

regularity that we choose for the kernel, and $\beta$ the prior we have on the optimum solution. For more details, we refer to section 4 of [22] or section 3.1 of [6].

## 4 Development of the SGD recursions and proof outline

In this section we provide an overview of the arguments that comprise the proof of our results.

**Recursions for multiplicative noise.** We can rewrite the SGD recursion Eq. (3) for the deviation to the optimum $\eta_t := \theta_t - \theta_*$ as a descent on the *risk* plus a multiplicative noise term. For $t \geqslant 1$,

$$\eta_{t+1} = \left(I - \gamma x_t x_t^\top\right) \eta_t = \left(I - \gamma H\right) \eta_t + \gamma \left(H - x_t x_t^\top\right) \eta_t. \tag{8}$$

We can also write the recursion satisfied by the covariance of the iterates, $M_t := \mathbb{E}\left[\eta_t \eta_t^\top\right]$:

$$M_{t+1} = \left(I - \gamma H\right) M_t \left(I - \gamma H\right) + \gamma^2 \mathbb{E}\left[\left(H - x_t x_t^\top\right) M_t \left(H - x_t x_t^\top\right)\right]. \tag{9}$$

Note a main difference between the two recursions, Eq. (9) is a deterministic recursion over operators whereas Eq. (8) is a stochastic recursion on vectors.

**Deviations sequence and initial conditions.** We emphasis that $\eta_t$ and $M_t$ represent *deviations* to the optimum $\theta_*$. This is why proving that $\theta_t$ goes to $\theta_*$ corresponds to $\eta_t$ and $M_t$ going to 0. As the initial point of the SGD recursion is $\theta_0 = 0$, the initial conditions $\eta_{t=0}$ and $M_{t=0}$ represent, in fact, respectively the optimum and the covariance at optimum.

**Eigenspaces of the covariance.** The core of the analysis rests on a reformulation of our problem in the eigenspaces of the covariance operator. Recall that $(\lambda_i, v_i)_i$ are the sorted eigenelements of $H$ and define the decomposition of $M_t$ in the basis of $v_i v_j^\top$, $M_t := \sum_i m_i^t v_i v_i^\top + \sum_{i \neq j} m_{ij}^t v_i v_j^\top$. We can write the expected risk of the estimator given by the $t$-th iterate of SGD, that we denote by $\mathsf{f}_t$:

$$\mathsf{f}_t := \mathbb{E}\mathcal{R}(\theta_t) = \frac{1}{2}\text{Tr}\left(M_t H\right) = \frac{1}{2}\sum_i \lambda_i m_i^t. \tag{10}$$

It is quite remarkable that the $(m_{ij})_{i \neq j}$ do not play any role in the recursions. The aim now is to write a recursion for $(m_i^t)_i$ that leads to a recursion for the function value $\mathsf{f}_t$ through Eq. (10). This is the purpose of the following and central lemma, whose proof can be found in Appendix 8.1.

**Lemma 4 (Recursion on the covariance of iterates)** *Define* $\mathsf{f}_i^t := \mathbb{E}\left[\langle v_i, X\rangle^2 X^\top M_t X\right]$.

*For $t \geqslant 1$, we have the following recursion on the covariance of the iterates of SGD, $\forall i$,*

$$m_i^{t+1} = m_i^t - 2\gamma \lambda_i m_i^t + \gamma^2 \mathsf{f}_i^t, \tag{11}$$

$$m_i^{t+1} = \left(1 - 2\gamma \lambda_i\right)^t m_i^0 + \gamma^2 \sum_{k=0}^{t} \left(1 - 2\gamma \lambda_i\right)^{t-k} \mathsf{f}_i^k, . \tag{12}$$

Let us make comments on this important lemma. First, note that the only difference with the deterministic recursion (gradient descent) is the presence of the "mixing terms" $(\mathsf{f}_i^t)_i$. In fact, these terms affect dramatically the dynamics. Indeed, there is no reason that the iterates decrease along the iterations: this is what makes the analysis more tedious for the final iterate. On the contrary, the usual convergence rate for the *averaged iterates* is easily obtained by summing Eq. (11) and comparing the terms. Note finally that the strength of Eq. (11) lies in the fact that the different eigenspaces (*e.g.* $m_i^t$ and $m_j^t$ for $i \neq j$) interact only through $\mathsf{f}_i^t$ and $\mathsf{f}_j^t$. By summing Eq. (12), and appropriately controlling sums of $\mathsf{f}_i^t$'s with Assumptions 1, 3 or 5, we can get recursive inequalities on $\mathsf{f}_t$. These show that $\mathsf{f}_t$ is in fact a Lyapunov function; going further, we use discrete versions of Gronwall-type inequalities to finally upper-bound it. To exemplify this reasoning, we present, in the following lemma, a Lyapunov control on $\mathsf{f}_t$ using Assumptions 1. Its proof can be found in Appendix 8.2.

**Lemma 5 (Recursion on $\mathsf{f}_t$)** *Under Assumption 1, assume $\gamma \leqslant \left(4\lambda_{\max}\right)^{-1}$. For $t \geqslant 1$ we have the following recursion on the function value $\mathsf{f}_t$. For all $i$,*

$$\mathsf{f}_t \leqslant \frac{\text{Tr}(M_0)}{4\gamma t} + \gamma\, R \sum_{k=0}^{t-1} \frac{\mathsf{f}_k}{t-k} \tag{13}$$

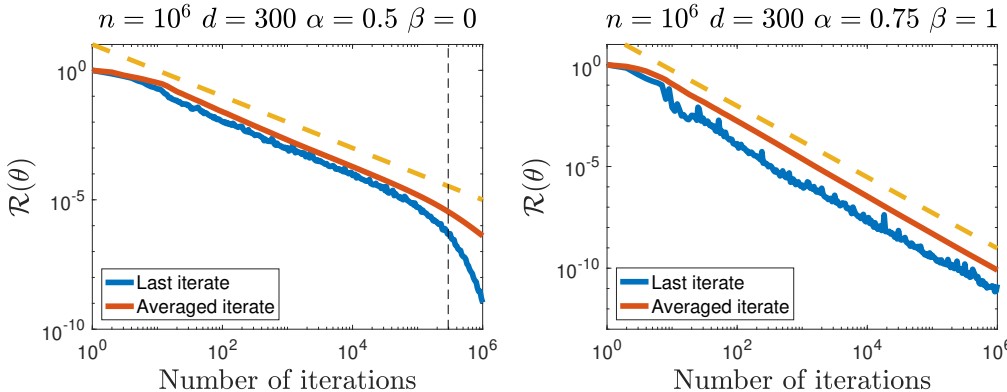

Figure 1: Least-squares regression. Left: $\alpha = 0.5$ and $\beta = 0$. The vertical dashed line marks the transition to the linear convergence regime. Right: $\alpha = 0.75$ and $\beta = 1$. The orange dashed line represents the curve $1/T^{1+\alpha\wedge\beta}$ predicted by Theorem 3.

The decrease of the function value $f_t$ is controlled by the sum of a bias term –characterizing how fast the initial conditions $m_i^0$ are forgotten–, and a variance term –characterizing how the noise reverberates through the iterates. All the theorems are proven using the same technique: the aim is to use the inequality recursively to control the variance term. The different Assumptions 1, 3, 5 lead to different variance term. Trying to factorize the proofs in one general bound does not allow for a clear presentation of the results. This is why, despite the apparent redundancy of the proof technique, for the sake of clarity and easy reading, we preferred to split the different proofs of the theorem and factorize only certain technical lemmas. All the proofs of Theorems 1, 2, 3 can be found in Appendix 9, respectively in Sections 9.1, 9.2, 9.3.

## 5 Experiments

We illustrate our theoretical results on a synthetic least-squares problem using the SGD algorithm defined in Eq. (3). For $d = 300$ we consider a stream of normally distributed inputs $(x_n)_n$ whose covariance matrix $H$ has random eigenvectors $v_i$ and eigenvalues $1/i^{1/(1-\alpha)}$ for $i = 1, \ldots, d$. The optimum is chosen randomly: $\theta_* = \sum 1/i^{\frac{1-\beta/(1-\alpha)}{2}} v_i$. This allows to reproduce the setting where the coefficient $\alpha$ and $\beta$ of the capacity and source conditions are perfectly controlled. The outputs $(y_n)_n$ are generated through $y_n = \langle \theta_*, x_n \rangle$. We take a step-size $\gamma = \frac{1}{2\mathrm{Tr}H}$. All results are averaged over ten repetitions. We compare the performance of the last iterate and the averaged iterate of SGD on two different problems: one corresponding to $\alpha = 0.5$ and $\beta = 0$ and the other to $\alpha = 0.75$ and $\beta = 1$. For those problems, the predicted convergence is given by Theorem 3: $O(1/T^{1+\alpha\wedge\beta})$. Hence we expect a $1/T$ convergence in the first example and a fast $1/T^{1.75}$ rates for the second.

First note that the bound of Theorem 3 is perfectly matched by these two examples. Second, we can see that, the averaged SGD and the final iterate show the same behavior quite accordingly to the theory (see comments on Theorem 3) even if we can notice a slight -but real- better performance (result are shown in $\log$-scale) from the final iterate. However, the main difference is that, as the averaged iterates show some saturation, the final iterate meets a point where it changes to a linear convergence regime (left plot in Figure 1). This adaptivity of the final iterate appearing after time scale $\tau \sim 1/(\gamma\lambda_{\min})$ represents a true asset in choosing the final iterate versus the averaged one. Finally notice that the linear rate regime cannot be seen in the right plot as the maximum number of iterations shown, $n = 10^6$, is negligible when compared to $\tau \sim d^{1/(1-\alpha)} = 300^{1/(1-0.75)} \sim 10^{10}$.

## 6 Conclusion and Perspectives

In this paper, we proceed to a detailed analysis of the convergence rates of SGD in the noiseless least-squares setting. We derive a systematic study of the SGD last iterate that leverages a sequence of fine-grained assumptions allowing us to exhibit fast polynomial rates. The absence of additive noise changes dramatically the behavior of SGD: no decaying step sizes or averaging are needed for

convergence as constant step-size SGD naturally adapts to the problem. The development of the SGD recursion as an interacting particle systems open many perspectives: can it be used to analyze SGD accelerations? multiple passes over the data? Another significant result is that, from an optimization perspective, we give a prototypal example where the last iterate of constant step-size SGD provably converges in the non-strongly convex case. This raises the following fundamental question: can we extend our method to show the convergence of SGD last iterate in the general non-strongly convex case?

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
