# Appendix

**Organization of the Appendix**

In Appendix 7 we present a detailed summary of our results and of the different assumptions. In Appendix 8 we prove Lemma 4 and 5 which enable to obtain the main SGD recursions. Then in Appendix 9 we prove our main results: Theorems 1, 2 and 3.

## 7 Detailed summary of the results and of the different assumptions

### 7.1 Capacity and Source Conditions and Results

**Summary on the assumptions.** All the assumptions we discussed in the article are summarized in Tables 2 and 3. It clearly shows that the assumptions go stricter and stricter revealing a finer description of the problem. Note also that they can be paired two by two, one sequence corresponding to the features and the other one to the optimum: Assumption 1 with 2, Assumption 3 with 4 and Assumption 5 with 6. The case $\beta \in (-1, 0)$, when the optimum is non-attainable, is left aside in the panels for the sake of clarity.

| Assumption | Name | Condition |
|---|---|---|
| Assumption **1** | Fourth Moment Condition | $\mathbb{E}\left[\|X\|^2 \, XX^\top\right] \preccurlyeq RH$ |
| $\Uparrow$ | | $\Uparrow$ |
| Assumption **3** | Features log-regularity | $\mathbb{E}\left[\langle X, \ln(H^{-1})X\rangle \, XX^\top\right] \preccurlyeq R_{\ln}H$ |
| $\Uparrow$ | | $\Uparrow$ |
| Assumption **5** | Capacity condition | $\mathbb{E}\left[\left\|H^{-\alpha/2}X\right\|^2 \, XX^\top\right] \preccurlyeq R_\alpha H, \alpha \in (0, 1)$ |

Table 2: Table showing conditions and implications for each Assumption on the features.

| Assumption | Name | Condition | Finite constant |
|---|---|---|---|
| Assumption **2** | Attainable case | $\|\theta_*\|_{\mathcal{H}} < +\infty$ | $\|\theta_*\|_{\mathcal{H}}$ |
| $\Uparrow$ | | $\Uparrow$ | |
| Assumption **4** | Optimum log-regularity | $\operatorname{Tr}(M_0 \ln(H^{-1})) < +\infty$ | $C_{\ln}$ |
| $\Uparrow$ | | $\Uparrow$ | |
| Assumption **6** | Source condition | $\operatorname{Tr}(M_0 H^{-\beta}) < +\infty, \ \beta > 0$ | $C_\beta$ |

Table 3: Table showing conditions and implications for each Assumption on the optimum.

**Summary on the results.** Each pair of assumptions (on the features and on the optimum) corresponds to a given rate of convergence. This is what is summarized in the Table 4. Stricter assumptions naturally lead to enhanced convergence rates or bigger step-sizes. Going from a theorem to the next one, the only difference is that we strengthen the assumptions to improve either the convergence rate or the step-size. Every theorem is a bound on the *expected risk* given by the *last iterate* of the SGD recursion with *constant* step size $\gamma$: $\mathbb{E}\mathcal{R}(\theta_T) = \frac{1}{2}\mathbb{E}\operatorname{Tr}\left[(\theta_T - \theta_*)(\theta_T - \theta_*)^\top H\right]$, where $\mathbb{E}$ stands for the expectation with respect to the stream of data used by the SGD algorithm until time $T$. Only for the Theorem 1, for which the assumptions are the weakest, we allow the step-size to depends on the finite horizon $T$ (through its logarithm). All the theorems are stated with respect to finite constants defined thanks to the different assumptions. The reader can refer to Tables 2 and 3 for a concise summary of them. All the proofs have been written in Appendix 9.

**Details on** $\log$**-regularity.** We have stated Assumption 3 with a reference $\lambda_\circ$ such that $\lambda_\circ \geq 7R_{\ln}$ for a better presentation of the results. Here we try to give a more detailed description on the choice of $\lambda_0$. Lets make an assumption without any dependence on $\lambda_\circ$: there exists a constant $\tilde{R}_{\ln} \geq 0$ such that:

$$\mathbb{E}\left[\langle X, \ln(H^{-1})X\rangle \, XX^\top\right] \preccurlyeq \tilde{R}_{\ln}H. \tag{14}$$

| Assumptions → Theorem | Rate | Step-size |
|---|---|---|
| Assumption **1**, **2** → Theorem **1** | $\ln(T)/T$ | $O(1/\ln(T))$ |
| ⇑ | | |
| Assumption **3**, **4** → Theorem **2** | $1/T$ | $O(1)$ |
| ⇑ | | |
| Assumption **5**, **6** → Theorem **3** | $1/T^{1+\alpha}$ | $O(1)$ |

Table 4: Table showing different upper-bounds for the convergence of the SGD final iterate given different assumptions.

Note that this assumption is stricter than Assumption 1 and hence there exists $\tilde{R} > 0$ such that $\mathbb{E}\left[\|X\|^2 X X^\top\right] \preccurlyeq \tilde{R}H$. Thus,

$$\mathbb{E}\left[\langle X, \ln\left(\lambda_{\mathrm{o}}H^{-1}\right)X\rangle X X^\top\right] = \mathbb{E}\left[\langle X, \ln\left(H^{-1}\right)X\rangle X X^\top\right] + \ln\lambda_{\mathrm{o}}\,\mathbb{E}\left[\|X\|^2 X X^\top\right]$$

$$\preccurlyeq \left(\tilde{R}\ln\lambda_{\mathrm{o}} + \tilde{R}_{\mathrm{ln}}\right)H.$$

Hence with $R_{\mathrm{ln}} = \tilde{R}\ln\lambda_{\mathrm{o}} + \tilde{R}_{\mathrm{ln}}$, it satisfies Assumption 3. Now coming to the condition, we need

$$7\left(\tilde{R}\ln\lambda_{\mathrm{o}} + \tilde{R}_{\mathrm{ln}}\right) \le \lambda_{\mathrm{o}}.$$

Without loss of generality assume $\tilde{R}_{\mathrm{ln}} \ge \tilde{R}$, $\ln\left(\ln\tilde{R}_{\mathrm{ln}}\right) \le \ln\tilde{R}_{\mathrm{ln}}$ and $\ln\tilde{R}_{\mathrm{ln}} \ge 1$. Let $\lambda_o = 50\tilde{R}_{\mathrm{ln}}\ln\tilde{R}_{\mathrm{ln}}$, then

$$7\left(\tilde{R}\ln\lambda_{\mathrm{o}} + \tilde{R}_{\mathrm{ln}}\right) \le 7\left(\tilde{R}_{\mathrm{ln}}\ln\left(50\tilde{R}_{\mathrm{ln}}\ln\tilde{R}_{\mathrm{ln}}\right)\right) + 7\tilde{R}_{\mathrm{ln}}$$

$$\le 7\ln 50\tilde{R}_{\mathrm{ln}} + 7\tilde{R}_{\mathrm{ln}}\ln\tilde{R}_{\mathrm{ln}} + 7\tilde{R}_{\mathrm{ln}}\ln\left(\ln\tilde{R}_{\mathrm{ln}}\right) + 7\tilde{R}_{\mathrm{ln}}$$

$$\le (7\ln 50 + 21)\tilde{R}_{\mathrm{ln}}\ln\tilde{R}_{\mathrm{ln}} \le 50\tilde{R}_{\mathrm{ln}}\ln\tilde{R}_{\mathrm{ln}} = \lambda_{\mathrm{o}}.$$

which satisfies the required condition. Hence we can always find such a $\lambda_{\mathrm{o}}$ if $\tilde{R}_{\mathrm{ln}}$ is finite.

**Capacity condition and eigenvalue decay.** In our comments on Assumptions 1,3,5, we make a few observations about the connection between the capacity conditions and the eigenvalue decay. Here, we give a detailed description between them. For example, let us assume Assumption 5,

$$\mathbb{E}\left[\langle X, H^{-\alpha}X\rangle X X^\top\right] \preccurlyeq R_\alpha H,$$

$$\mathrm{Tr}\left(\mathbb{E}\left[\langle X, H^{-\alpha}X\rangle X X^\top H^{-\alpha}\right]\right) \le \mathrm{Tr}\left(R_\alpha H^{1-\alpha}\right), \qquad \text{as } H \text{ is PSD}$$

$$\mathbb{E}\left[\langle X, H^{-\alpha}X\rangle\mathrm{Tr}(X X^\top H^{-\alpha})\right] \le R_\alpha\mathrm{Tr}\left(H^{1-\alpha}\right), \qquad \text{using } \mathrm{Tr}(X X^\top H^{-\alpha}) = \langle X, H^{-\alpha}X\rangle,$$

$$\mathbb{E}\left[\langle X, H^{-\alpha}X\rangle^2\right] \le R_\alpha\mathrm{Tr}\left(H^{1-\alpha}\right).$$

Now using Cauchy-Schwarz we have,

$$\left(\mathrm{Tr}H^{1-\alpha}\right)^2 = \mathrm{Tr}\left(\mathbb{E}\left[X X^\top\right]H^{-\alpha}\right)^2 = \mathbb{E}\left[\langle X, H^{-\alpha}X\rangle\right]^2 \le \mathbb{E}\left[\langle X, H^{-\alpha}X\rangle^2\right] \le R_\alpha\mathrm{Tr}\left(H^{1-\alpha}\right).$$

So from Assumption 5 we have $\mathrm{Tr}H^{1-\alpha} \le R_\alpha$. Using the fact that the eigen values are enumerated in sorted non-increasing order, we have

$$i\lambda_i^{1-\alpha} \le \sum_{j\le i}\lambda_j^{1-\alpha} \le \mathrm{Tr}H^{1-\alpha} \le R_\alpha$$

which gives that $\lambda_i$ is of order $O(1/i^{\frac{1}{1-\alpha}})$.

Similarly, if we do similar analysis for Assumption 3, it implies that $\mathrm{Tr}\left(H\ln(\lambda_0 H^{-1})\right) \le R_{\mathrm{ln}}$ and hence, $\lambda_i$ is of order $O(\frac{1}{i\ln i})$.

# 8 Proofs of the SGD recursions: Lemma 4 and 5

## 8.1 Proof of Lemma 4

First recall that for $t \geqslant 1$, Eq. (9) gives that

$$M_{t+1} = (I - \gamma H) M_t (I - \gamma H) + \gamma^2 \mathbb{E}\left[\left(H - x_t x_t^\top\right) M_t \left(H - x_t x_t^\top\right)\right].$$

Lets compute $\mathbb{E}\left[\left(H - x_t x_t^\top\right) M_t \left(H - x_t x_t^\top\right)\right]$. Using that $\mathbb{E}[x_t x_t^\top] = H$, we first write,

$$\mathbb{E}\left[\left(H - x_t x_t^\top\right) M_t \left(H - x_t x_t^\top\right)\right] = \mathbb{E}\left[H M_t H\right] - \mathbb{E}\left[x_t x_t^\top M_t H\right] - \mathbb{E}\left[H M_t x_t\, x_t^\top\right] + \mathbb{E}\left[x_t x_t^\top M_t\, x_t x_t^\top\right]$$

$$= H M_t H - H M_t H - H M_t H + \mathbb{E}\left[x_t^\top M_t x_t\, x_t x_t^\top\right]$$

$$= \mathbb{E}\left[x_t^\top M_t x_t\, x_t x_t^\top\right] - H M_t H.$$

Now using this in Eq. (9) we have

$$M_{t+1} = (I - \gamma H) M_t (I - \gamma H) + \gamma^2 \left[\mathbb{E}\left[x_t^\top M_t x_t\, x_t x_t^\top\right] - H M_t H\right]$$

$$= M_t - \gamma H M_t - \gamma M_t H + \gamma^2 \mathbb{E}\left[x_t^\top M_t x_t\, x_t x_t^\top\right].$$

Now we project the above term on $v_i v_i^\top$

$$v_i^\top M_{t+1} v_i = v_i^\top M_t v_i - \gamma v_i^\top H M_t v_i - \gamma v_i^\top M_t H v_i + \gamma^2 v_i^\top \mathbb{E}\left[x_t^\top M_t x_t\, x_t x_t^\top\right] v_i$$

$$= v_i^\top M_t v_i - 2\gamma \lambda_i v_i^\top M_t v_i + \gamma^2 \mathbb{E}\left[\langle v_i, x_t\rangle^2\, x_t^\top M_t x_t\right]. \qquad \text{Using } H v_i = \lambda_i v_i.$$

Hence, using the notation $\mathsf{f}_i^t = \mathbb{E}\left[\langle v_i, x_t\rangle^2\, x_t^\top M_t x_t\right]$, and recalling that $v_i^\top M_t v_i^\top = m_i^t$, we have the following recursion,

$$m_i^{t+1} = m_i^t - 2\gamma \lambda_i m_i^t + \gamma^2 \mathbb{E}\left[\langle v_i, x_t\rangle^2\, x_t^\top M_t x_t\right].$$

This proves the first part of Lemma 4.

For the second part, we prove the recursion Eq. (12) using an induction argument. For the base case $t = 1$ we can see that the Lemma holds directly from Eq. (11). Assume that the Lemma holds for $k = t - 1$. We know from Eq. (11) that

$$m_i^t = (1 - 2\gamma \lambda_i) m_i^{t-1} + \gamma^2 \mathsf{f}_i^t,$$

and from the induction hypothesis, we have

$$m_i^{t-1} = (1 - 2\gamma \lambda_i)^{t-1} m_i^0 + \gamma^2 \sum_{k=0}^{t-2} \lambda_i (1 - 2\gamma \lambda_i)^{t-k-2} \mathsf{f}_i^k.$$

Merging these above two equalities we have,

$$m_i^t = (1 - 2\gamma \lambda_i) \left((1 - 2\gamma \lambda_i)^{t-1} m_i^0 + \gamma^2 \sum_{k=0}^{t-2} \lambda_i (1 - 2\gamma \lambda_i)^{t-k-2} \mathsf{f}_i^k\right) + \gamma^2 \lambda_i \mathsf{f}_i^k$$

$$= (1 - 2\gamma \lambda_i)^t m_i^0 + \gamma^2 \sum_{k=0}^{t-1} \lambda_i (1 - 2\gamma \lambda_i)^{t-k-1} \mathsf{f}_i^k.$$

This completes the proof of Lemma 4.

## 8.2 Proof of Lemma 5

Here we prove the Lemma 5. First note that from Eq. (12) of Lemma 4 we have the following recursion,

$$m_i^t = (1 - 2\gamma \lambda_i)^t m_i^0 + \gamma^2 \sum_{k=0}^{t-1} (1 - 2\gamma \lambda_i)^{t-k-1} \mathsf{f}_i^k. \tag{15}$$

We take $\gamma \leq (4\lambda_{\max})^{-1}$ to get a cleaner version of the recursion. It can be easily verified that the maximal learning rate specified in each of the theorems will satisfy this. We have,

$$\lambda_i m_i^t \leq \lambda_i \left(1 - 2\gamma\lambda_i\right)^t m_i^0 + 2\gamma^2 \sum_{k=0}^{t-1} \lambda_i \left(1 - 2\gamma\lambda_i\right)^{t-k} \mathsf{f}_i^k. \tag{16}$$

For $x \in (0, 1)$, we have $x(1-x)^t \leq xe^{-xt} \leq 1/t$. Now using the following natural bounds, we get

$$\lambda_i(1 - 2\gamma\lambda_i)^{t-k} \leqslant \frac{1}{2\gamma(t-k)}$$

$$\lambda_i m_i^t \leq \frac{m_i^0}{2\gamma t} + \gamma \sum_{k=0}^{t-1} \frac{\mathsf{f}_i^k}{t-k}.$$

Hence,

$$2\mathsf{f}_t = \sum_i \lambda_i m_i^t \leqslant \sum_i \frac{m_i^0}{2\gamma t} + \gamma \sum_{k=0}^{t-1} \left(\sum_i \mathsf{f}_i^k\right) \frac{1}{t-k}. \tag{17}$$

Now lets compute the sum below. Recalling $\mathsf{f}_i^t = \mathbb{E}\left[\langle v_i, x_t\rangle^2 x_t^\top M_t x_t\right]$, we have

$$\begin{aligned}
\sum_i \mathsf{f}_i^t &= \sum_i \mathbb{E}\left[\langle v_i, x_t\rangle^2 x_t^\top M_t x_t\right] \\
&= \mathbb{E}\left[\left(\sum_i \langle v_i, x_t\rangle^2\right) x_t^\top M_t x_t\right] \\
&= \mathbb{E}\left[\|x_t\|^2 \operatorname{Tr}\left(x_t x_t^\top M_t\right)\right] \\
&= \operatorname{Tr}\left[\mathbb{E}\left[\|x_t\|^2 x_t x_t^\top\right] M_t\right].
\end{aligned}$$

Note for any two PSD matrices $A, B$ if $A \preccurlyeq B$ then $\operatorname{Tr}(AM) \leq \operatorname{Tr}(BM)$, for any PSD matrix $M$. From Assumption 1, $\mathbb{E}\left[\|x_t\|^2 x_t x_t^\top\right] \preccurlyeq RH$ and we thus have $\sum_i \mathsf{f}_i^t \leqslant R\operatorname{Tr}(M_t H) = 2R\mathsf{f}_t$. Now replacing the sum in Eq. (17), we get finally,

$$\mathsf{f}_t \leq \frac{\operatorname{Tr}(M_0)}{4\gamma t} + \gamma\,R \sum_{k=0}^{t-1} \frac{\mathsf{f}_k}{t-k}.$$

This proves Lemma 5.

# 9 Proofs of the main results

## 9.1 Proof of Theorem 1

From Lemma 5, we have for all $t \geqslant 1$,

$$\mathsf{f}_t \leqslant \frac{\operatorname{Tr}(M_0)}{4\gamma t} + \gamma R \sum_{k=0}^{t-1} \frac{\mathsf{f}_k}{t-k}.$$

And then, the technique we use throughout all the proofs of this section rest on a control of the second term $\sum_{k=0}^{t-1} \frac{f_k}{(t-k)}$ using the recursion. Indeed, by summing,

$$\sum_{t=0}^{T-1} \frac{f_t}{T-t} \leqslant \frac{f_0}{T} + \frac{\mathrm{Tr}(M_0)}{4\gamma} \sum_{t=1}^{T-1} \frac{1}{t(T-t)} + \gamma R \sum_{t=1}^{T-1} \sum_{k=0}^{t-1} \frac{f_k}{(t-k)(T-t)}$$

$$\leqslant \frac{f_0}{T} + \frac{\mathrm{Tr}(M_0)}{4\gamma} \sum_{t=1}^{T-1} \frac{1}{t(T-t)} + \gamma R \sum_{t=1}^{T-1} \sum_{k=0}^{t-1} \frac{f_k}{(t-k)(T-t)}$$

$$= \frac{f_0}{T} + \frac{\mathrm{Tr}(M_0)}{4\gamma T} \sum_{t=1}^{T-1} \left[\frac{1}{t} + \frac{1}{T-t}\right] + \gamma R \sum_{k=0}^{T-2} f_k \sum_{t=k+1}^{T-1} \frac{1}{(t-k)(T-t)}$$

$$\leqslant \frac{f_0}{T} + \frac{\mathrm{Tr}(M_0)\ln(T)}{2\gamma T} + 2\gamma R\ln(T) \sum_{k=0}^{T-1} \frac{f_k}{T-k}.$$

Noting that $f_0 = \sum_i \lambda_i m_i^0 \leqslant \lambda_{\max} \mathrm{Tr}(M_0) \leqslant \frac{\mathrm{Tr}(M_0)}{2\gamma}$, for $T \geqslant 2$,

$$\sum_{t=0}^{T-1} \frac{f_t}{T-t} \leqslant \frac{\mathrm{Tr}(M_0)\ln(T)}{\gamma T} + 2\gamma R\ln(T) \sum_{k=0}^{T-1} \frac{f_k}{T-k}$$

$$(1 - 2\gamma R\ln(T)) \sum_{t=0}^{T-1} \frac{f_t}{T-t} \leqslant \frac{\mathrm{Tr}(M_0)\ln(T)}{\gamma T}.$$

Hence, for $\gamma = (4R\ln(T))^{-1}$, we have

$$\sum_{t=0}^{T-1} \frac{f_t}{T-t} \leqslant \frac{2\mathrm{Tr}(M_0)\ln(T)}{\gamma T},$$

and, hence, for all $T \geqslant 2$,

$$f_T \leqslant \frac{\mathrm{Tr}(M_0)}{4\gamma T} + \gamma R \sum_{k=0}^{T-1} \frac{f_k}{T-k}$$

$$\leqslant \frac{R\mathrm{Tr}(M_0)\ln(T)}{T} + \frac{2R\mathrm{Tr}(M_0)\ln(T)}{T}$$

$$\leqslant 3R\mathrm{Tr}(M_0)\frac{\ln(T)}{T}.$$

That concludes the proof of Theorem 1.

## 9.2 Proof of Theorem 2

The proof of this theorem follows the same principle but uses slightly better estimations. Indeed, we replace the previous $1/n$ bound by a finer bound. This is the statement of the following lemma. For this, we define the following summation $S_n(x)$ for some $x \in (0, 1/4]$ and $n \geq 2$,

$$S_n(x) := \sum_{k=0}^{n-1} \frac{(1-x)^k}{n-k}. \tag{18}$$

**Lemma 6** *For any $x \in (0, 1/4]$, $n \geq 1$, we have the following bound*

$$xS(x) \leqslant \frac{7\ln(1/x)}{n}. \tag{19}$$

**Proof**

In the first step, we slightly reformulate this expression, then try to bound that expression by a continuous integral which gives the desired result. From Eq. (18), we have

$$S_n(x) = \sum_{k=0}^{n-1} \frac{(1-x)^k}{n-k} = (1-x)^n \sum_{k=0}^{n-1} \frac{(1-x)^{k-n}}{n-k}$$

$$(1-x)^{-n} S_n(x) = \sum_{k=1}^{n} \frac{(1-x)^{-k}}{k}$$

We compute $(1-x)^{-n} S_n(x)$. Note that $g(y) = (1-x)^{-y}/y$ is convex for $y > 0$. For any convex function $g$, we have that for any integer $k \geq 2$, $y \in [k, k+1]$ we have that $g(k) \leq g(y) + g(y-1)$. Indeed, if $g$ is in the increasing phase then $g(y)$ dominates $g(k)$, else $g(y-1)$ dominates in the decreasing phase. Using this we can see that for $k = 2 \cdots n$,

$$g(k) \leq \int_k^{k+1} (g(y) + g(y-1))\, dy$$

$$\sum_{k=2}^{n-1} \frac{(1-x)^{-k}}{k} \leq \int_2^n (g(y) + g(y-1))\, dy.$$

Hence,

$$(1-x)^{-n} S_n(x) = \sum_{k=1}^{n} \frac{(1-x)^{-k}}{k} \leq (1-x)^{-1} + 2 \int_1^n \frac{(1-x)^{-y}}{y} + \frac{(1-x)^{-n}}{n}\, dy. \qquad (20)$$

This leads us to try to bound the integral $\int_1^{n+1} \frac{(1-x)^{-y}}{y}$. Using the change of variable $(1-x)^{-y} = e^t$. We can rewrite the above integral as follows:

$$\int_1^n \frac{(1-x)^{-y}}{y}\, dy = \int_{-\ln(1-x)}^{-n\ln(1-x)} \frac{e^t}{t}\, dt.$$

For the sake of clearer notations, let us define $a := -\ln(1-x)$ such that as $0 < x < 1$, we have $a > 0$. The equation simplifies to the following

$$\int_1^n \frac{(1-x)^{-y}}{y} \leq \int_a^{na} \frac{e^t}{t}\, dt$$

$$\leq \int_a^1 \frac{e^t}{t}\, dt + \int_1^{na} \frac{e^t}{t}\, dt$$

For $t \leq 1$, we can see that $e^t/t \leq e/t$, where as for $t \geq 2$ we use the bound $e^t/t \leq 2e^t(t-1)/t^2$. Using these bounds,

$$\int_1^n \frac{(1-x)^{-y}}{y} \leq e \int_a^1 \frac{1}{t}\, dt + \int_1^2 \frac{e^t}{t} + 2 \int_2^{na} \frac{(t-1)e^t}{t^2}\, dt$$

$$\leq e \ln\frac{1}{a} + e^2 - e + 2 \left[\frac{e^x}{x}\right]_2^{na}$$

$$\leq e \ln\frac{1}{a} + e^2 - e + 2 \left[\frac{e^{na}}{na} - \frac{e^2}{2}\right]$$

$$\leq e \ln\frac{1}{a} + 2\frac{e^{na}}{na}$$

Re-substituting, $a = -\ln(1-x)$, we can see the $a \geqslant x$ and hence,

$$\int_1^n \frac{(1-x)^{-y}}{y} \leq e \ln\frac{1}{x} + 2\frac{(1-x)^{-n}}{nx},$$

such that we can write

$$(1-x)^{-n} S_n(x) \le (1-x)^{-1} + 2 \left[ e \ln \frac{1}{x} + 2 \frac{(1-x)^{-n}}{nx} \right] + \frac{(1-x)^{-n}}{n}$$

$$S_n(x) \le (1-x)^{n-1} + 2e \ln \frac{1}{x} (1-x)^{-n} + \frac{4}{n} + \frac{1}{n}$$

$$x S_n(x) \le x(1-x)^{n-1} + 2e \; x(1-x)^{-n} \; \ln \frac{1}{x} + \frac{4}{n} + \frac{x}{n}.$$

Now we use the fact that $x(1-x)^n \le xe^{-nx} \le 1/(en)$. Leveraging also that $x \le 1/4$, so that $\ln \frac{1}{x} > 1$ and $(1-x)^{-1} \le 4/3$. We can finally bound,

$$x S_n(x) \le \frac{4}{3e} \frac{1}{n} + \frac{1}{4n} + \frac{4}{n} + 2\frac{1}{n} \ln \frac{1}{x} \le \frac{7}{n} \ln \frac{1}{x}.$$

This completes the proof.

Using the above we prove Theorem 2. Recall that for all $t \ge 1$, from Eq.(12) and $\gamma \le (4\lambda_{max})^{-1}$

$$m_i^t \le (1 - 2\gamma\lambda_i)^t \, m_i^0 + 2\gamma^2 \sum_{k=0}^{t-1} (1 - 2\gamma\lambda_i)^{t-k} \, \mathsf{f}_i^k.$$

The technique we use in this section rests on a control of the second term using the recursion. For this, we will use carefully the bound in Lemma 6. As said before, the only difference with the proof of the previous theorem is the special care in estimations to avoid the logarithm at the price of a slightly more stringent assumption. Indeed, $\forall i$,

$$\sum_{t=0}^{T-1} \frac{m_i^t}{T-t} = \sum_{t=0}^{T-1} \frac{(1 - 2\gamma\lambda_i)^t}{T-t} m_i^0 + 2\gamma^2 \sum_{t=0}^{T-1} \sum_{k=0}^{t-1} (1 - 2\gamma\lambda_i)^{t-k} \, \mathsf{f}_i^k$$

$$\le \frac{m_i^0}{T} + \sum_{t=1}^{T-1} \frac{(1 - 2\gamma\lambda_i)^t}{T-t} m_i^0 + 2\gamma^2 \sum_{k=0}^{T-2} \mathsf{f}_i^k \sum_{t=k+1}^{T-1} \frac{(1 - 2\gamma\lambda_i)^{t-k}}{T-t}$$

$$\sum_{t=0}^{T-1} \frac{\lambda_i m_i^t}{T-t} \le \frac{\lambda_i m_i^0}{T} + \sum_{t=1}^{T-1} \frac{(1 - 2\gamma\lambda_i)^t}{T-t} m_i^0 + \gamma \sum_{k=0}^{T-2} \mathsf{f}_i^k \sum_{t=1}^{T-k-1} (2\gamma\lambda_i) \frac{(1 - 2\gamma\lambda_i)^t}{T-k-t}$$

$$\le \frac{\lambda_i m_i^0}{T} + \frac{m_i^0}{2\gamma} (2\gamma\lambda_i) S_T(2\gamma\lambda_i) + \gamma \sum_{k=0}^{T-1} \mathsf{f}_i^k \, (2\gamma\lambda_i) S_{T-k}(2\gamma\lambda_i).$$

And then, by applying Lemma 6 :

$$\sum_{t=0}^{T-1} \frac{\lambda_i m_i^t}{T-t} \le \frac{\lambda_i m_i^0}{T} + \frac{7m_i^0}{2\gamma} \ln \left( \frac{1}{2\gamma\lambda_i} \right) \frac{1}{T} + 7\gamma \sum_{k=0}^{T-1} \mathsf{f}_i^k \, \ln \left( \frac{1}{2\gamma\lambda_i} \right) \frac{1}{T-k}.$$

For now lets assume that $\lambda_{\mathrm{o}}$ be such that $2\gamma\lambda_{\mathrm{o}} \ge 1$, we will check this later at the end. We have,

$$\sum_{t=0}^{T-1} \frac{\lambda_i m_i^t}{T-t} \le \frac{\lambda_i m_i^0}{T} + \frac{7m_i^0}{2\gamma} \ln \left( \frac{\lambda_{\mathrm{o}}}{\lambda_i} \right) \frac{1}{T} + 7\gamma \sum_{k=0}^{T-1} \mathsf{f}_i^k \, \ln \left( \frac{\lambda_{\mathrm{o}}}{\lambda_i} \right) \frac{1}{T-k}.$$

Now summing over $i$, we get

$$\sum_{t=0}^{T-1} \sum_i \frac{\lambda_i m_i^t}{T-t} \le \sum_i \frac{\lambda_i m_i^0}{T} + \sum_i \frac{7m_i^0}{2\gamma} \ln \left( \frac{\lambda_{\mathrm{o}}}{\lambda_i} \right) \frac{1}{T} + 7\gamma \sum_{k=0}^{T-1} \left( \sum_i \mathsf{f}_i^k \ln \left( \frac{\lambda_{\mathrm{o}}}{\lambda_i} \right) \right) \frac{1}{T-k}. \quad (21)$$

Note from Eq.(10), we have $\sum_i \lambda_i m_i^t = 2\mathsf{f}_t$. Lets calculate the remaining terms

$$\sum_i \mathsf{f}_i^t \, \ln \left( \frac{\lambda_{\mathrm{o}}}{\lambda_i} \right) = \sum_i \ln \left( \frac{\lambda_{\mathrm{o}}}{\lambda_i} \right) \mathbb{E} \left[ \left( \langle v_i, x_t \rangle^2 \right) x_t^\top M_t x_t \right]$$

$$= \mathbb{E} \left[ \left( \sum_i \ln \left( \frac{\lambda_{\mathrm{o}}}{\lambda_i} \right) \langle v_i, x_t \rangle^2 \right) x_t^\top M_t x_t \right]$$

$$= \mathbb{E} \left[ \langle x_t, \ln \left( \lambda_{\mathrm{o}} H^{-1} \right) x_t \rangle \, \mathrm{Tr} \left( x_t x_t^\top M_t \right) \right]$$

$$= \mathrm{Tr} \left[ \mathbb{E} \left[ \langle x_t, \ln \left( \lambda_{\mathrm{o}} H^{-1} \right) x_t \rangle \, x_t x_t^\top \right] M_t \right].$$

Hence, as in the previous case, using Assumption 3, we have $\sum_i f_i^t \ln\left(\frac{\lambda_o}{\lambda_i}\right) \leqslant 2R_{\ln} f_t$. From Assumption 4, we recognize the second term of Eq. (21), $\sum_i m_i^0 \ln\left(\frac{\lambda_o}{\lambda_i}\right) = \mathrm{Tr}\left(M_0 \ln\left(\lambda_o H^{-1}\right)\right) = C_{\ln}$. Substituting all that in Eq. (21) we get,

$$\sum_{t=0}^{T-1} \frac{f_t}{T-t} \leq \frac{f_0}{T} + \frac{7C_{\ln}}{4T} + 7\gamma R_{\ln} \sum_{t=0}^{T-1} \frac{f_t}{T-t}.$$

And like in the previous proof, $f_0 = \sum_i \lambda_i m_i^0 \leqslant \lambda_{\max} \sum_i m_i^0 \leqslant \frac{1}{\gamma} \sum_i m_i^0 \ln(1/(\gamma \lambda_i)) \leq C_{\ln}/\gamma$.

$$\sum_{t=0}^{T-1} \frac{f_t}{T-t} \leqslant \frac{3C_{\ln}}{\gamma T} + 7\gamma R_{\ln} \sum_{k=0}^{T-1} \frac{f_k}{T-k}.$$

Hence, for $\gamma = (14R_{\ln})^{-1}$, we have

$$\sum_{t=0}^{T-1} \frac{f_t}{T-t} \leqslant \frac{6C_{\ln}}{\gamma T},$$

and, hence, we conclude like for the previous theorems. We know that Assumption 3 is stricter than Assumption 1 i.e there exists a constant $R' = \left(\ln\left(\frac{\lambda_o}{\lambda_{max}}\right)\right)^{-1} R_{\ln}$ such that Assumption 1 holds with this. Indeed, this allows us to use Lemma 5, for all $T \geqslant 1$,

$$\begin{aligned}
f_t &\leqslant \frac{\mathrm{Tr}(M_0)}{4\gamma T} + \gamma R' \sum_{k=0}^{T-1} \frac{f_k}{T-k} \\
&\leqslant \frac{14R_{\ln}\mathrm{Tr}(M_0)}{4T} + \frac{6R' C_{\ln}}{T} \\
&\leqslant \frac{4R_{\ln}\mathrm{Tr}(M_0)}{T} + \frac{6R' C_{\ln}}{T}.
\end{aligned}$$

We can always choose $\lambda_o$ large enough such that $\left(\ln\left(\frac{\lambda_o}{\lambda_{max}}\right)\right) > 1$. With this we note that $\mathrm{Tr}(M_0) \leqslant C_{\ln}$ and $\mathrm{Tr}(H) \leqslant R_{\ln}$. Hence,

$$f_t \leqslant \frac{10R_{\ln}C_{\ln}}{T}.$$

That concludes the proof of Theorem 2.

Note that we have $2\lambda_o \gamma \geqslant 1$ for $\gamma = (14R_{\ln})^{-1}$, as we chose $\lambda_o$ such that $7R_{\ln} \leqslant \lambda_o$.

### 9.3 Proof of Theorem 3

Once again we proceed with the same technique as above. The aim here is to tighten the estimation for both the first and the second term with the capacity and source assumptions of the problem (Assumptions 5 and 6).

This estimation rests on the inequality stated in the following Lemma.

**Lemma 7** *For $x \in (0,1)$ and $t \geqslant 1$, for $r > 0$ we have the following inequality*

$$x^r(1-x)^t \leq \frac{r^r}{t^r}. \tag{22}$$

**Proof** Let $x \in (0,1)$, $t \geqslant 1$ and $r > 0$. It is standard to note that $(1-x)^t \leqslant e^{-tx}$. Hence,

$$x^r(1-x)^t \leqslant x^r e^{-tx}.$$

Now, a rapid look at the maximum of the function $x \to x^r e^{-tx}$ gives us that it attains its maximum for $x = r/t$. Hence

$$x^r (1-x)^t \leqslant x^r e^{-tx} \leqslant \left(\frac{r}{t}\right)^r e^{-r} \leqslant \frac{r^r}{t^r},$$

and this proves the Lemma.

Again, recall that for all $t \geqslant 1$, from Eq.(12) and $\gamma \leq (4\lambda_{max})^{-1}$, $\forall i$,

$$m_i^t \leqslant (1 - 2\gamma\lambda_i)^t m_i^0 + 2\gamma^2 \sum_{k=0}^{t-1} (1 - 2\gamma\lambda_i)^{t-k} f_i^k,$$

$$\lambda_i m_i^t \leqslant \lambda_i^{1+\beta} (1 - 2\gamma\lambda_i)^t \lambda_i^{-\beta} m_i^0 + 2\gamma^2 \lambda_i^{-\alpha} \sum_{k=0}^{t-1} \lambda^{1+\alpha} (1 - 2\gamma\lambda_i)^{t-k} f_i^k.$$

Thanks to Lemma 7, we can bound the above expression as,

$$\lambda_i m_i^t \leqslant \left(\frac{1+\beta}{2\gamma t}\right)^{1+\beta} \lambda_i^{-\beta} m_i^0 + 2\gamma^2 \sum_{k=0}^{t-1} \left(\frac{1+\alpha}{2\gamma t}\right)^{1+\alpha} \lambda_i^{-\alpha} f_i^k.$$

Summing across $i$'s we get

$$\sum_i \lambda_i m_i^t \leqslant \left(\frac{1+\beta}{2\gamma t}\right)^{1+\beta} \sum_i \lambda_i^{-\beta} m_i^0 + 2^{-\alpha} \gamma^{1-\alpha} (1+\alpha)^{1+\alpha} \sum_{k=0}^{t-1} \frac{1}{t^{1+\alpha}} \sum_i \lambda_i^{-\alpha} f_i^k.$$

Note from Eq.(10), we have $\sum_i \lambda_i m_i^t = 2f_t$. Lets calculate the remaining terms

$$\sum_i f_i^t \lambda_i^{-\alpha} = \sum_i \lambda_i^{-\alpha} \mathbb{E}\left[\left(\langle v_i, x_t\rangle^2\right) x_t^\top M_t x_t\right]$$

$$= \mathbb{E}\left[\left(\sum_i \lambda_i^{-\alpha} \langle v_i, x_t\rangle^2\right) x_t^\top M_t x_t\right]$$

$$= \mathbb{E}\left[\langle x_t, H^{1-\alpha} x_t\rangle \operatorname{Tr}(x_t x_t^\top M_t)\right]$$

$$= \operatorname{Tr}\left[\mathbb{E}\left[\langle x_t, H^{1-\alpha} x_t\rangle x_t x_t^\top\right] M_t\right].$$

From Assumption 5, we thus have, $\sum_i f_i^t \lambda_i^{-\alpha} \leqslant 2R_\alpha f_t$. And we get from Assumption 6 that $\sum_i m_i^0 \lambda_i^{-\beta} = \operatorname{Tr}(M_0 H^{-\beta}) = C_\beta$. Substituting the above computed terms, we get the following

$$f_t \leqslant \frac{C_\beta}{2}\left(\frac{1+\beta}{2\gamma t}\right)^{1+\beta} + 2^{-\alpha} (1+\alpha)^{1+\alpha} \gamma^{1-\alpha} R_\alpha \sum_{k=0}^{t-1} \frac{f_k}{(t-k)^{1+\alpha}}. \tag{23}$$

And from this, we use the same technique as for the previous theorems to bound the second term of the right side of the inequality. To accomplish this we need a bound on the following sum. For $\beta > -1, \alpha \in (0,1), T \geqslant 2$,

$$S_T(\alpha, \beta) := \sum_{t=1}^{T-1} \frac{1}{t^{1+\beta}(T-t)^{1+\alpha}}$$

This is the aim of the Lemma below.

**Lemma 8** *For $\beta > -1$, $\alpha \in (0,1)$, $T \geqslant 2$, we have the following upper-bound,*

$$S_T(\alpha, \beta) \leqslant \frac{2^{2+\alpha\wedge\beta}\xi_{\alpha\vee\beta}}{T^{1+\alpha\wedge\beta}},$$

*where, for $u > 0$, $\xi_u := \sum_{k\geqslant 1} \frac{1}{k^{1+u}}$ and we use the following classical notations: $\alpha\wedge\beta = \min(\alpha, \beta)$ and $\alpha\vee\beta = \max(\alpha, \beta) > 0$.*

**Proof** Let us assume that $\beta \leqslant \alpha$, we have,

$$
\begin{aligned}
\mathsf{S}_T(\alpha, \beta) = \sum_{t=1}^{T-1} \frac{1}{t^{1+\beta}(T-t)^{1+\alpha}} &= \sum_{t=1}^{T-1} \frac{1}{(t(T-t))^{1+\beta}} \frac{1}{(T-t)^{\alpha-\beta}} \\
&= \frac{1}{T^{1+\beta}} \left[ \sum_{t=1}^{T-1} \frac{1}{(T-t)^{\alpha-\beta}} \left( \frac{1}{t} + \frac{1}{T-t} \right)^{1+\beta} \right] \\
&= \frac{2^{1+\beta}}{T^{1+\beta}} \left[ \sum_{t=1}^{T-1} \frac{1}{(T-t)^{\alpha-\beta} t^{1+\beta}} + \sum_{t=1}^{T-1} \frac{1}{(T-t)^{\alpha+1}} \right].
\end{aligned}
$$

Now the second term is trivially upper bounded by $\xi_\alpha$. And for the first term, we use Young's inequality with coefficient $(p, q) = (\frac{1+\alpha}{\alpha-\beta}, \frac{1+\alpha}{1+\beta})$:

$$
\begin{aligned}
\sum_{t=1}^{T-1} \frac{1}{(T-t)^{\alpha-\beta} t^{1+\beta}} &\leqslant \frac{1}{p} \sum_{t=1}^{T-1} \frac{1}{(T-t)^{(\alpha-\beta)p}} + \frac{1}{q} \sum_{t=1}^{T-1} \frac{1}{t^{(1+\beta)q}} \\
&= \frac{1}{p} \sum_{t=1}^{T-1} \frac{1}{t^{1+\alpha}} + \frac{1}{q} \sum_{t=1}^{T-1} \frac{1}{t^{1+\alpha}} \\
&= \sum_{t=1}^{T-1} \frac{1}{t^{1+\alpha}} \leqslant \xi_\alpha.
\end{aligned}
$$

This concludes the proof is the case where $\beta \leqslant \alpha$.

Symmetrically, if $\alpha \leqslant \beta$, by a change of variable $t \to T - t$,

$$
\mathsf{S}_T(\alpha, \beta) = \sum_{t=1}^{T-1} \frac{1}{t^{1+\beta}(T-t)^{1+\alpha}} = \sum_{t=1}^{T-1} \frac{1}{t^{1+\alpha}(T-t)^{1+\beta}} = \mathsf{S}_T(\beta, \alpha),
$$

and the proof follows.

Thanks to the Lemma we continue the proof of Theorem 3. Recall Eq. (23): for $T \geqslant 2$,

$$
\mathsf{f}_t \leqslant \frac{C_\beta}{2} \left( \frac{1+\beta}{2\gamma t} \right)^{1+\beta} + 2 \left( \frac{1+\alpha}{2} \right)^{1+\alpha} \gamma^{1-\alpha} R_\alpha \sum_{k=0}^{t-1} \frac{\mathsf{f}_k}{(t-k)^{1+\alpha}}.
$$

Hence, we proceed like previously,

$$
\begin{aligned}
\sum_{t=0}^{T-1} \frac{\mathsf{f}_t}{(T-t)^{1+\alpha}} \leqslant {}&\frac{\mathsf{f}_0}{T^{1+\alpha}} + \frac{C_\beta}{2} \left( \frac{1+\beta}{2\gamma} \right)^{1+\beta} \sum_{t=1}^{T-1} \frac{1}{t^{1+\beta}(T-t)^{1+\alpha}} \\
&+ 2^{-\alpha}(1+\alpha)^{1+\alpha} \gamma^{1-\alpha} R_\alpha \sum_{t=1}^{T-1} \sum_{k=0}^{t-1} \frac{\mathsf{f}_k}{(t-k)^{1+\alpha}(T-t)^{1+\alpha}} \\
\leqslant {}&\frac{\mathsf{f}_0}{T^{1+\alpha}} + \frac{C_\beta}{2} \left( \frac{1+\beta}{2\gamma} \right)^{1+\beta} \mathsf{S}_T(\alpha, \beta) \\
&+ 2^{-\alpha}(1+\alpha)^{1+\alpha} \gamma^{1-\alpha} R_\alpha \sum_{k=1}^{T-1} \mathsf{f}_k \sum_{t=k+1}^{T-1} \frac{1}{(t-k)^{1+\alpha}(T-t)^{1+\alpha}} \\
\leqslant {}&\frac{\mathsf{f}_0}{T^{1+\alpha}} + \frac{C_\beta}{2} \left( \frac{1+\beta}{2\gamma} \right)^{1+\beta} \mathsf{S}_T(\alpha, \beta) + 2^{-\alpha}(1+\alpha)^{1+\alpha} \gamma^{1-\alpha} R_\alpha \sum_{k=1}^{T-1} \mathsf{f}_k \mathsf{S}_{T-k}(\alpha, \alpha).
\end{aligned}
$$

And applying Lemma 8, and the fact that $\frac{f_0}{T^{1+\alpha}} \leqslant \frac{C_\beta}{\gamma^\beta T^{1+\alpha \wedge \beta}}$,

$$\sum_{t=0}^{T-1} \frac{f_t}{(T-t)^{1+\alpha}} \leqslant \frac{f_0}{T^{1+\alpha}} + \frac{C_\beta}{2} \left(\frac{1+\beta}{2\gamma}\right)^{1+\beta} \frac{2^{2+\alpha \wedge \beta} \xi_{\alpha \vee \beta}}{T^{1+\alpha \wedge \beta}}$$

$$+ 2^{-\alpha}(1+\alpha)^{1+\alpha}\gamma^{1-\alpha}R_\alpha 2^{2+\alpha}\xi_\alpha \sum_{k=1}^{T-1} \frac{f_k}{(T-t)^{1+\alpha}}$$

$$\leqslant 2C_\beta \left(\frac{1+\beta}{\gamma}\right)^{1+\beta} \frac{\xi_{\alpha \vee \beta}}{T^{1+\alpha \wedge \beta}}$$

$$+ 4(1+\alpha)^{1+\alpha}\xi_\alpha \gamma^{1-\alpha} R_\alpha \sum_{k=1}^{T-1} \frac{f_k}{(T-t)^{1+\alpha}}.$$

Now, for $\gamma$ such that $4(1+\alpha)^{1+\alpha}\xi_\alpha \gamma^{1-\alpha} R_\alpha \leqslant 1/2$, i.e., for simplicity,

$$\gamma^{1-\alpha} \leqslant (32\xi_\alpha R_\alpha)^{-1},$$

then, we have,

$$\sum_{t=0}^{T-1} \frac{f_t}{(T-t)^{1+\alpha}} \leqslant 4C_\beta \left(\frac{1+\beta}{\gamma}\right)^{1+\beta} \frac{\xi_{\alpha \vee \beta}}{T^{1+\alpha \wedge \beta}},$$

and, hence, we conclude like for the previous theorems. Indeed, for all $T \geqslant 1$, recalling Eq. (23): for $T \geqslant 2$,

$$f_T \leqslant \frac{C_\beta}{2} \left(\frac{1+\beta}{2\gamma T}\right)^{1+\beta} + 2^{-\alpha}(1+\alpha)^{1+\alpha}\gamma^{1-\alpha}R_\alpha \sum_{k=0}^{T-1} \frac{f_t}{(T-t)^{1+\alpha}}$$

$$\leqslant \frac{C_\beta}{2} \left(\frac{1+\beta}{2\gamma T}\right)^{1+\beta} + \frac{1}{8\xi_\alpha} \sum_{t=0}^{T-1} \frac{f_t}{(T-t)^{1+\alpha}}$$

$$\leqslant \frac{C_\beta}{2} \left(\frac{1+\beta}{2\gamma T}\right)^{1+\beta} + \frac{1}{2\xi_\alpha} C_\beta \left(\frac{1+\beta}{\gamma}\right)^{1+\beta} \frac{\xi_{\alpha \vee \beta}}{T^{1+\alpha \wedge \beta}}$$

$$\leqslant \frac{C_\beta}{2} \left(\frac{1+\beta}{2\gamma T}\right)^{1+\beta} + \frac{C_\beta}{2} \left(\frac{1+\beta}{\gamma}\right)^{1+\beta} \frac{1}{T^{1+\alpha \wedge \beta}}$$

$$\leqslant 2C_\beta \left(\frac{1+\beta}{\gamma}\right)^{1+\beta} \frac{1}{T^{1+\alpha \wedge \beta}}.$$

That concludes the proof of Theorem 3.