# OpenReview forum: "Last iterate convergence of SGD for Least-Squares in the Interpolation regime."
_NeurIPS.cc/2021/Conference — NeurIPS 2021 Poster_

### Official Review · Reviewer_Mkow · 2021-06-29

**Rating:** 6
**Confidence:** 4

**Summary:**

The paper explores the last iterate convergence of SGD on the least squares problem in the interpolation regime (noiseless setting). The authors work in an abstract Hilbert space and thus consider (1) the non-strongly convex setting and (2) the streaming or online setting. There are three contributions of the paper:

(1). The final iterate of constant step-size SGD achieves a convergence rate of $O(ln(T)/T)$ under the assumption that the data matrix $X$ has (in expectation) bounded fourth moments.

(2). Under more assumption about the behavior of the eigenvalues of the covariance matrix of the data, that is $\lambda_i \le O(1/i \log(i))$, the authors can improve the last iterate convergence rate to $O(1/T)$. This matches the rate for the averaged iterates of SGD.

(3). Under stronger assumptions of the eigenvalues of the covariance matrix at the optimum (source) and capacity condition, one can get an even faster rate of convergence.

**Limitations And Societal Impact:**

The limitations of the results are clearly stated. This result only applies to the least-squares problem and in the streaming setting. The results are purely theoretically.

**Main Review:**

The paper is well written and the details of the exact assumptions are clearly stated and there are enough details to follow the main ideas of the results. Although I did not check the proofs in detail, I am confident that they are correct. One concern about this paper is that it is fully devoted to trying to remove a log factor, which I understand is of interest theoretically, but of practical interest one may not even see. It is not entirely clear that one could even remove the log factor in the general setting. Can one show in simulated data that the $1/T$ is achieved?

I would also suggest that the authors elucidate on the $\alpha$ and $\beta$. While I understand that these assumptions have been used in previous works, it is not clear what are "natural choices" for these parameters. For instance in the case of $X$ having i.i.d. Gaussian entries of mean $0,1$ and $\theta^*$ pure noise, I believe one could compute the values of $\alpha$ and $\beta$. Could the authors comment on this?

There is also some recent work on last iterate convergence in the multi-pass setting (see Paquette, et al. "SGD in the Large") and could the authors comment on comparisons with this work (to match the streaming setting $r = \infty$)?





**Time Spent Reviewing:**

3

---

> ### Author Response · Authors · 2021-08-10
> **Review answer**
>
> Thank you very much for the time you took to review this paper.
>
> **Removing the log factor, tight rates and $\alpha, \beta$ conditions.** We kindly think that there is a confusion here regarding our results and our motivations ; i.e., the paper *is not fully devoted* to remove the log-factor.
> -  The result in Theorem 1 is novel and non-trivial. Therefore, it is a contribution per se, even with the additional log factor.
> -  If Theorem 2 can be viewed as an effort to remove the extra log factor of Theorem 1 under additional assumption, the rates derived under the source and capacity conditions in Theorem 3 are of independent interest. They indeed follow a different goal: the aim is to give *tight rates* with classical fine grained assumptions on the problem (see [3] for the result that the given rates are optimal). To detail a bit on these assumptions, for a broader audience, the capacity and source conditions (Assumption 5, 6) can be seen as *a priori* on the difficulty of the problem defined through the covariance matrix and the complexity of the optimal predictor. They are connected to classes of regularity of the problem in the non-parametric and kernel communities [Page 5 of [1]]. Note that these assumptions are not only usual in these communities but also an important measure of the performance of a given estimator. In your example of Gaussian data and $\theta_*$ pure noise: the answer is that $\alpha = +\infty$ (covariance matrix eigenvalues decrease as exponential), and $\beta$ would depend on the model class of the noise: this is the purpose of an entire field named *Gaussian processes* [2].
>
> **Link with Paquette et al. work**. Thank you for pointing out this nice and interesting reference. We will include it in our revised related work section. Nevertheless, there is a strong difference between our two approaches: our bounds are non-asymptotic, corresponding to $d = \infty$ and finite $n$ whereas in their work the results are asymptotic and both $n,d \to \infty$. Yet, when $r=1$ their problem is also not strongly convex and it is very interesting to see that they obtain similar rates: this can be seen when comparing our Theorem 3 (with $\beta$ arbitrarily large) to Paquette et. al Theorem E.3 with $r = 1$ and $\widetilde{R} = 0$.
>
> $ $
>
> [1] *Introduction to Nonparametric Estimation*, Tsybachov, 2003, Springer.
>
> [2] *Gaussian processes in Machine Learning*, Rasmussen, 2003, Springer.
>
> [3]  R. Berthier, F. Bach, P. Gaillard. *Tight Nonparametric Convergence Rates for Stochastic Gradient Descent under the Noiseless Linear Model*. NeurIPS '20 - 34th International Conference on Neural Information Processing Systems, Dec 2020.

---

### Official Review · Reviewer_ttkk · 2021-07-15

**Rating:** 6
**Confidence:** 3

**Summary:**

This paper studies the problem of minimizing a non-strongly convex Least-Squares using stochastic gradient descent (SGD). The authors show that when the interpolation assumption holds, then under mild assumptions, the last iterate of SGD converge to the optimal solution with rate $\mathcal{O}(\frac{\log(T)}{T})$. And under stronger assumptions, the convergence rate can be improved to $\mathcal{O}(\frac{1}{T})$.

**Limitations And Societal Impact:**

There is no negative societal impact.

**Main Review:**

I found the manuscript to be clearly written and technically sound.  However, there are still a few points need further clarification:
1. One important setting in this paper is that the covariance operator $H = \mathbb{E}[X \times X]$ is not positive definite, namely the least square is not strongly convex. However, in Assumption 1, it assumes that $H \succeq \frac{\mathbb{E}[\|X\|^2 XX^T]}{R}$. Does it mean that the covariance operator is actually positive definite? Same question for Assumption 3.
2. In the discussion after Assumption 4, you have the condition $\lambda_{\min} > 0$. I think this is also equivalent to assuming the problem is strongly convex.

Based on the discussion above, I think although there is some novelty in the problem setting and the corresponding theoretical result. The contribution may not be enough provided there are already some result about the convergence of last iterate from the SGD on strongly convex functions.


**Time Spent Reviewing:**

2 hrs

---

> ### Author Response · Authors · 2021-08-10
> **Review answer**
>
> Thank you very much for the time you took to review this paper. It is very puzzling that you are thinking that our work corresponds to the strongly convex case (see a detailed answer to your concerns below). This precise fact is commented throughout the paper: indeed, the lure of strong convexity is discussed in detail in l.140 and empirically validated l.346. Hence, we politely disagree with the claim that our contribution is incremental given previous works for strongly convex functions.
>
> **Assumption $H \geq \mathbb{E}[\|X\|^2XX^\top] $ and Assumptions 3, 5.** The first Assumption does not imply that the operator is positive definite: indeed you can imagine that when the data is normalised, e.g. $\|X\|^2 = 1$, the assumption would simply become $H \geq H$. Hence, it does not imply that $H\geq \mu I$. For Assumption 3 or 5, it is quite the same: as noted below the assumptions, it is only equivalent to assume that the eigenvalue decay of $H$ is faster than a power law (or some log-refinement of it), so there may not exists $\mu$ such that $H\geq \mu I$: $0$ may be a accumulative point of the spectrum of $H$.
>
> **Assumption 4 and strong convexity.** Note that in l.204, it is written « **when** the spectrum of the covariance matrix is lower bounded... ». Here we just wanted to give an order of magnitude in the special case of finite dimension. But **we do not assume that $\lambda_{min} > 0$ in any case** ($\lambda_{min}$ is not even defined properly in the infinite dimension case). Moreover, as discussed in the paper, in the finite dimensional case, even if the covariance matrix is positive definite, and the problem is, strictly speaking, strongly convex, the smallest eigenvalue can be arbitrarily small, therefore non-strongly convex assumptions better capture this setting. For example, if the smallest eigenvalue of $H$ if of order $10^{-7}$ while the trace stays constant $\mathrm{tr}H=1$, existing convergence results for the last iterate of SGD on strongly convex functions ensure a rate $O(e^{-10^{-7}T})$ after T iterations whereas the rate given in our paper is $O(\frac{\ln{T}}{T})$. As a consequence, for finite horizons (say at least for $10^{7}$ iterations) the guarantee on the convergence rate provided by our work will considerably surpass the rates given by previous results for strongly convex functions. We will make this fact even clearer: this is absolutely crucial to understand our work and there should be no confusion.

---

> > ### Comment · Reviewer_ttkk · 2021-08-20
> > **Response to author**
> >
> > Thank you for the explanation. The score is increased.

---

### Official Review · Reviewer_cbwB · 2021-07-16

**Rating:** 6
**Confidence:** 2

**Summary:**

This paper studies the least-squares problem in the interpolation regime, solved by the last iterate of the Stochastic Gradient Descent. It characterizes the performance of SGD under 3 sets of assumptions and shows the explicit impact of the conditions on the convergence rate.

**Limitations And Societal Impact:**

Not applicable due to theoretical character of the paper.

**Main Review:**

I am not very well familiar with the literature on the infinite-dimensional least-squares problem. My background is mainly in the field of convex optimization, so I will make the comments mainly from this perspective.
Firstly, the assumptions seem strange and hardly interpretable, and I think that the paper can benefit from drawing connections to general convex stochastic optimization. At least it would be helpful for non-experts in this area because, for now, it seems targeted at a very narrow audience.

Could you please tell me what is the purpose of Assumption 2, if the noiseless problem statement includes the existence of $\theta_* \in \mathcal{H}$ such that $\<\theta_*, x> = y$.

How can you use the $H^{-1}$ in Assumption 3 and further, if on page 3 (line 131) it is said that *“$H$ is no longer invertible”*?

The paper is clearly written, and the “related work” section seems complete. Though, excessive details ultimately make it harder to get the broad picture, distinguish the paper's contributions, and compare it to the previous results (especially in terms of technical novelty).

To be honest, I do not quite get the point of considering the infinite-dimensional case, and the motivation presented by the authors does not seem convincing to me. That is why it is hard to see if the results are valuable for the machine learning community.

From the practical viewpoint, the paper seems quite irrelevant, as I can not see any helpful insights for practitioners. The synthetic experiments are very basic, although this is clearly not the main focus of the work.

**Time Spent Reviewing:**

7

---

> ### Author Response · Authors · 2021-08-10
> **Review answer**
>
> Thank you very much for the time you took to review this paper.
>
> **Linear model, link with convex optimisation and model assumptions.** We understand the reviewer’s concern about the links between our assumptions and the traditional ones used in convex optimisation. To summarise these, let us stress that all these assumptions are related to the covariance operator $H$. Indeed, the resulting function to optimise is smooth (in the sense of Liptschitzness of derivatives) with the smoothness parameter being the largest eigenvalue $\lambda_{max}(H)$. And it is strongly convex with parameter $\lambda_{min}(H)$ and only convex when $\lambda_{min}(H) = 0$. Note that with this linear modelling, it is known in the kernel community that the convergence rates of the optimisation procedures can be described at a finer grain: this is done through quantifying how fast the eigenvalue of $H$ go to zero (Assumptions 3 or 5) and how aligned the target solution $\theta_*$ is with the covariance matrix eigenvectors (Assumption 4 or 6). We will clarify this classical setup to broaden the understanding of our analysis to the larger Neurips community.
>
> **Purpose of Assumption 2 in the interpolation model.** This is a truly fair concern and we thank the reviewer for raising this subtle fact. In a first reading, Assumption 2 is only here to put emphasis on that even if $\theta_*$ belongs to $\mathcal{H}$, its norm will play a crucial role in the convergence rate of Theorem 1. However, to tackle the unattainable case when the norm is infinite, we need to define $ y = \langle \theta_*,x\rangle_{\mathcal{H}}$ even when $\theta_*$ belongs to *larger* spaces than $\mathcal{H}$. The subtle fact is that even though $\theta_*$ may not belong to $\mathcal{H}$, we still can define $\langle\theta_*, x\rangle_{\mathcal{H}} x = y x $ through an extended covariance operator. For this construction, we refer to the note in Appendix B.1.4 of [6]. We will add this for the sake of completeness in our own Appendix.
>
> **Use of $H^{-1}$ despite non-invertibility.** First, let us give an important comment that we will add in the revised version: we can always assume that $H$ has no eigenvalue $0$ by projecting the data onto the minimal subspace where the inputs almost surely lie (refer to the comment after A3 in [7]). Then, for the sake of clarity let us distinguish between two cases:
> - *Finite dimensional case*:  in this case, the spectrum is indeed lower bounded, but with a value that can be arbitrarily small. This is what we explain in the paragraph **The impossibility of linear rates**. In this case, it is clear how to define functions of $H^{-1}$.
> - *Infinite dimensional case*: in this case, the spectrum is *not* lower bounded as $0$ is an accumulative point of the spectrum ($H$ being compact). Yet, even if $H^{-1}$ is not properly defined as a continuous operator, $H$ is a Positive Semi Definite compact operator that can be decomposed into an (infinite) eigenbasis. Unbounded operator theory [3] allows us to define functions of $H$ as acting directly on their spectral decomposition, that is for $H = \sum_i \lambda_i u_i u_i^\top$, we can always make proper operations with $f(H) = \sum_i f(\lambda_i) u_i u_i^\top$. Applying this to $f: x \to x^{-1}$ makes our assumptions theoretically valid.
>
> **Motivation for the infinite dimensional case.** We are convinced that the infinite dimensional case is very interesting to grasp the flavours of the current setup of practical machine learning (extremely large dimensionality of today’s tasks). Furthermore, the infinite dimensional case appears naturally when considering a non-parametric family of estimators, i.e., kernel methods, [4]. In fact, it is worth noting that there is a renewal in the interest on kernel methods as they can be used to analyse deep neural networks learning through the Neural Tangent Kernel or Random Fourier Features [1,2]. In this abundant literature, the feature map $\phi(x)$ is often of very large dimension (and sometimes infinite) and prevents the use of adaptive non-linear methods: this explains why kernel methods are largely used in practice, even beyond the machine learning community [5]. To improve clarity, we will highlight the link with this literature.
>
> $ $
>
> [1] Jacot, A., Gabriel, F. and Hongler, C., 2018. *Neural tangent kernel: Convergence and generalization in neural networks.* arXiv preprint arXiv:1806.07572.
>
> [2] A. Rahimi and B. Recht. 2007. *Random features for large-scale kernel machines.* In Proceedings of the 20th International Conference on Neural Information Processing Systems 2007.
>
> [3] *Unbounded self-adjoint operators on Hilbert space,* K. Schmudgen, Springer, 2012.
>
> [4] *Introduction to Nonparametric Estimation*, Tsybakov, 2003, Springer.
>
> [5] *Kernel techniques: from machine learning to meshless methods*, Schaback, 2006, Acta Numerica.
>
> [6] A. Dieuleveut, F. Bach. *Nonparametric stochastic approximation with large step-sizes.* The Annals of Statistics, 44(4) 1363-1399 August 2016.
>
> [7] F. Bach and E. Moulines. 2013. *Non-strongly-convex smooth stochastic approximation with convergence rate O(1/n)*. In Proceedings of the 26th International Conference on Neural Information Processing Systems.

---

> > ### Comment · Reviewer_cbwB · 2021-08-19
> > **Clarification question**
> >
> > As soon as you said that the considered case is not strongly convex: so $\lambda_{min}(H) = 0$, then I still have a question about how the operator $H^{-1}$ is defined in this case? The way suggested by you is not fully clear for me, specifically how $f: x \to x^{-1}$ should be defined for $\lambda_i = \lambda_{min}(H) = 0$?
> >
> > I am curious if it is possible to investigate how the inaccurate choice of step-size affects your results to understand better the applicability to the real world? It would be interesting to understand if your bounds on the step-size are tight: e.g. larger ones will not guarantee such rates of convergence.
> >
> > Connections to multiple passes seem also very interesting.
> >
> > It would be helpful to give a more detailed explanation of why the result in Theorem 1 is *novel and non-trivial*. I think that a broader audience will benefit from it.

---

> > > ### Author Response · Authors · 2021-08-23
> > > **Answer to questions**
> > >
> > > **Question:** *As soon as you said that the considered case is not strongly convex: so $\lambda_{min}(H) = 0$, then I still have a question about how the operator $H^{-1}$  is defined in this case? The way suggested by you is not fully clear for me, specifically how $f: x \to x^{-1}$ should be defined for $\lambda_i = \lambda_{min}(H) = 0$ ?*
> > >
> > >
> > > **Answer:** As explained in our rebuttal (paragraph *Use of $H^{-1}$ despite non-invertibility*), we are assuming that $0$ is not an eigenvalue of the covariance operator $H$. *Hence, we never state that $\lambda_{min}=0$*. Note that such an assumption is not restrictive since it is always possible to project the data onto the minimal subspace where the inputs almost surely lie.
> > >
> > > As already said, the finite dimensional case is not problematic as $\lambda_{min}$ is properly defined and strictly positive (but possibly arbitrarily small). Surely, the infinite dimensional case is more involved:  there we have a infinite sequence $\left(\lambda_i\right)$ of eigenvalue converging towards $0$: $\lambda_i \underset{i \to \infty}{\to} 0$. That is even if $\forall i \geq 1$, $\lambda_i > 0$, the problem is not strongly convex as the spectrum is not lower bounded by a strictly positive constant! Note that $\lambda_{min}$ is even not properly defined here as the infinite sequence decreases to $0$. With this in mind, the definition of $H^{-1}$ through its spectrum decomposition makes sense as we never apply $f : x \to x^{-1}$ in $0$.
> > >
> > > For the sake of clarity, let's give a precise example for the infinite dimensional case: Let us consider that $H$ is the infinite diagonal matrix $[\mathrm{diag}(\frac{1}{i^2})]$ acting on vectors of $\mathbb{R}^{\mathbb{N}}$. Of course its eigenvectors are the infinite canonical basis $(e_i)$, where $e_i$ is the vector of $\mathbb{R}^{\mathbb{N}}$ with a $1$ in the $i$-th position and $0$ otherwise. Then, we can define $H^{-1}$, as explained in the rebuttal, as the diagonal matrix  $[\mathrm{diag}(i^2)]$. Note however that even if this defines properly a linear operator of $\mathbb{R}^{\mathbb{N}}$, it is not continuous as it is not bounded (for linear operator, boundedness and continuity are equivalent): $\| H^{-1}\|_{op} = \mathrm{sup}_i \lambda_i =  \mathrm{sup}_i i^2 = + \infty $. Yet, we can define operations with respect to it: for example for $v = (\frac{1}{i^2})_i \in \mathbb{R}^{\mathbb{N}}$, $H^{-1} v = \boldsymbol{1} $, where $\boldsymbol{1} \in \mathbb{R}^{\mathbb{N}}$ is the vector with $1$ at every coordinates.
> > >
> > > **Question:** *I am curious if it is possible to investigate how the inaccurate choice of step-size affects your results to understand better the applicability to the real world? It would be interesting to understand if your bounds on the step-size are tight: e.g. larger ones will not guarantee such rates of convergence.*
> > >
> > > **Answer:** Thank you for the very interesting question. Our bounds on the step-size are not perfectly tight, but Theorem 2 and 3 show step-size of the correct order with respect to optimal step-sizes $O(2/Tr(H))$ (see discussion page 3, 2nd column of  Defossez et al, 2015).
> > >
> > > **Question:** *Connections to multiple passes seem also very interesting.*
> > >
> > > **Answer:** The connection with multiple passes is indeed very interesting. We can tackle this setting when considering results on the training loss. However, deriving results for the population error is far more challenging (due to autocorrelations of the samples) and connection with benign overfitting would certainly be needed. It is an exciting direction for future research.
> > >
> > > **Question:** *It would be helpful to give a more detailed explanation of why the result in Theorem 1 is novel and non-trivial. I think that a broader audience will benefit from it.*
> > >
> > > **Answer:** Thank you for this comment which has not been addressed in our rebuttal. We will simplify the related work in order to make the novelty of our results clearer. To summarize, no previous results on the last iterate convergence of constant-step-size SGD for non-strongly convex functions were known. The existing results were either heavily using some variance reduction technique (decreasing step-sizes or averaging), or corresponds to the strongly convex setting. In a quite novel way, we prove the convergence of the last iterate of **constant-step-size** SGD, despite the multiplicative noise present in the gradient estimates.
> > > In terms of technical relevance: as explained in [l.314-l.317], the function value sequence is non-decreasing and no Lyapunov function can be easily found, unlike in the strongly convex case. In fact, we have to control the fluctuations caused by the SGD noise without using classical variance reduction techniques (e.g., averaging or decreasing step-size). Therefore we have to resort to explicitly handling the variance by direct expansion. Accordingly, we think our proof technique is non trivial, novel and of independent interest.

---

### Official Review · Reviewer_vd3S · 2021-07-16

**Rating:** 6
**Confidence:** 4

**Summary:**

This paper investigates the excess risk bounds of constant-stepsize SGD last iterate in the setting of noise-less least squares (i.e., only considering the bias error). The presented bounds could be applied to overparameterized cases, but are also limited to attainable cases. The presented bounds for the last iterate nearly match the established ones for the averaged iterates.

**Limitations And Societal Impact:**

See above.

**Main Review:**

# Pros:
+ The presented bounds for the last iterates of SGD are the first results that can be applied in the overparameterized setting.
+ Some of the presented bounds are (nearly) tight and (partly) solve an open question for the rate of SGD last iterate.
+ Lemma 5 is interesting in its own for bounding the SGD (bias) iterates.
+ Overall the paper is organized well, but the writing could be improved here and there.

# Cons:
- The considered least square problems are noiseless, i.e., the variance error bound is not considered. In fact I am thinking that using a large and constant stepsize and without averaging, the variance term cannot be decreased. In this sense the presented results and techniques are somehow less satisfactory.
- The presented bound, e.g., Thm1, requires the true parameter to have a finite $\ell_2$-norm (attainable case), which is limited and could potentially be relaxed to an even more interesting version like $(\theta^*)^\top H \theta^* < \infty$.
- The writing could be improved in several places. See below.

# Small issues:
* The paper/section/paragraph title capitalization style should be made consistent.
* l.7. two folds.
* l.25. This sentence is quite confusing. Note that a noisy model could also interpolate. See e.g., [bartlett et al 2020].
* l.37. To be more clear: "... a decreasing sequence in terms of loss"
* l.47. an infinite..
* l.48. This is also confusing, as strongly convex cases are covered by the presented results. Better to write, e.g., "the problem we are considering is not restricted to strongly convex or finite-dim cases".
* l.49. the following two.
* l.55. Everything is known for noisy setting (which covers noiseless setting)? Perhaps you mean the finite-dim setting?
* l.68. This is too confusing... Lots of papers listed in paragraph l.76 explicitly or implicitly discussed how to control SGD last iterate. I agree this paper has some new results for SGD last iterate but the claim here is too exaggerative!
* related works section is quite messy. Perhaps the authors could discuss papers for offline/multi-pass SGD and online/one-pass SGD separately? These two are quite different in terms of generalization bounds. Also, there are big paper overlaps in the three paragraphs & could find better ways to organize.
* l.144. an arbitrarily...
* l.149. Not only in large dim case, but in any case.
* l.168. a 4th...
* l.173. at least some results in [10] only assume eq (4) instead of something stronger.
* l.203. $C_{ln}$ is not defined. I guess it refers to left hand side of the above inequality?
* l.227. an eigenvalue...
* l.321. remove the empty space before ?

# Overall
Based on my knowledge on papers for SGD last iterate, I think the presented results have their spirits and could be a good reading for controlling SGD last iterate. I would love to see further discussions/results to relax the finite $\ell_2$-norm (of the true parameter) condition, as the $\ell_2$-norm easily blows up in infinite-dim cases. Discussions on how to control the variance error could also be useful. As for the writing, there are plenty places need to revised though. My current evaluation for this paper is neutral and slightly lean to weak acceptance.



[bartlett et al 2020] Bartlett, Peter L., et al. "Benign overfitting in linear regression." Proceedings of the National Academy of Sciences 117.48 (2020): 30063-30070.

**Time Spent Reviewing:**

5

---

> ### Author Response · Authors · 2021-08-10
> **Review answer**
>
> Thank you very much for the time you took to read and comment on this paper. We are very grateful for your work that will help us a lot in clarifying, and improving the paper.
>
> **Relaxed assumptions on the norm of $\theta_*$.** We totally agree that the norm of the optimal predictor is a crucial quantity (as it controls the convergence rate of many statistical estimators). In the non-parametric setting, e.g., kernel methods, this norm could even be infinite as wisely noted. In Theorem 1, we wanted to present the *most basic bound* for the sake of clarity: this is why we assume that the norm of $\theta_*$ is bounded (attainable case). Going further, as it is classically done in the non-parametric literature, we present bounds in Theorem 3 where **the norm of $\theta_*$ is not assumed finite** hence we deal with the non-attainable case the reviewer is concerned with. More precisely the parameter $\beta$ introduced in Assumption 6 models our *a priori* on $\theta_*$ and $\beta \in (-1,0)$ corresponds to unattainable cases. Additionally, note that this case is rarely dealt with in the non-parametric community. We will make this important fact clearer as it is a major contribution of the presented paper.
>
> **Dealing with noise.** As noted, if the model has noise (with intensity $\sigma^2$) an additional variance has to be considered ; yet we could derive a similar analysis and change our results adding a variance term $\gamma R \sigma^2$ (as it is done in Theorem 5 in [1]). Thus, as noted by the reviewer, the iterates are not converging anymore to the solution, but *around* the solution. This explains that in that setting, variance reduction techniques (decreasing step-size or averaging) are preferred. Note that for this precise least-square problem with noise, averaged SGD is optimal [2].
>
> **Motivation to study final iterate SGD in the noiseless setting.** The successes of overparameterized models in machine learning has led the community to consider the noiseless case where the outputs do not suffer from any noise. Note however that there is still some noise in the procedure in the *intrinsic sampling noise of SGD*. In this noiseless case, final iterate SGD should not suffer with such an asymptotic variance (see previous paragraph); this is why resorting to variance reduction strategies (e.g, decaying step sizes or averaging techniques) should not be necessary. This is the main motivation of our work. We will motivate this aspect more carefully and compare more precisely to the related work; we will also tone down our claim of l.68 in the final version in order to make our contribution clearer.
>
>
> **Note on finite number of data and interpolation.** As mentioned thoroughly by the reviewer, observations following a noisy linear model can also be interpolated. Our framework naturally adapts to this finite sum setting, considering uniform sampling with replacement on a finite-size data set. We will also make this fact clearer and relate to the appropriate literature.
>
> **Typos.** Thank you for the typos and the notes on the imprecisions in the writing: we will take them into consideration and this will help a lot with the clearness of the paper.
>
> $ $
>
> [1] R. Berthier, F. Bach, P. Gaillard. *Tight Nonparametric Convergence Rates for Stochastic Gradient Descent under the Noiseless Linear Model.* NeurIPS '20 - 34th International Conference on Neural Information Processing Systems, Dec 2020.
>
> [2]  A. Dieuleveut, F. Bach. *Nonparametric stochastic approximation with large step-sizes.* The Annals of Statistics, 44(4) 1363-1399 August 2016.

---

> > ### Comment · Reviewer_vd3S · 2021-08-13
> > **Reply to the authors**
> >
> > Thanks for the reply.
> >
> > * According to the authors' reply, we all agree that the presented techniques for constant-stepsize SGD cannot deal with addictive noise / variance error (in the sense of obtaining a decreasing excess risk). While this is regretful to hear, I do not think this diminishes the spirit of the presented techniques for bounding the bias error, and I strongly encourage the authors to explicitly discuss this in their revision to benefit future readers.
> >
> > * This claim confuses me: "The successes of overparameterized models in machine learning has led the community to consider the noiseless case where the outputs do not suffer from any noise."
> > I could understand that overparameterized model motivates the research of the interpolation regime (that a model is capable to perfectly fit all observed data). I am not sure in which sense this motivates the research of no-addictive-noise setting. Note that overparameterization permits an alternative model assumption; why this could motivate a new data distribution assumption? Perhaps the authors should seek for a different motivation for the studied no-addictive-noise setting.
> >
> >
> > The above limitations prevents me from increasing the score for this paper. I am still slightly lean to weak acceptance, but rather neutral.

---

> > > ### Author Response · Authors · 2021-08-17
> > > **Answer to Comment**
> > >
> > > We thank the reviewer for the reply and the positive assessment. However we are worried there might still be some misunderstandings which we would like to clarify.
> > >
> > > ## Overparameterization: assumptions on the models/on the data
> > >
> > > **Question:** *I could understand that overparameterized model motivates the research of the interpolation regime (that a model is capable to perfectly fit all observed data). I am not sure in which sense this motivates the research of no-additive-noise setting.*
> > >
> > > **Answer:** In the interpolation regime, where it exists an interpolator $\theta_*$ which perfectly fits the training data $(x_i, y_i)$, i.e., $\ell(f_{\theta_*}(x_i),y_i)=0$ for $i \leq n$, each gradient of the individual loss is zero, i.e.,  $\nabla_\theta \ell(f_{\theta_*}(x_i),y_i)=0$. Indeed, each loss is positive and therefore is minimized in $\theta_*$. Thus the variance of the SGD gradient estimator $\nabla_\theta \ell(f_{\theta}(x_i),y_i)$, for $i$ uniformly sampled in $[1,n]$ can be bounded as $E_i  \| \nabla \ell(f_{\theta}(x_i),y_i) \|^2 = E_i \| \nabla \ell(f_{\theta}(x_i),y_i)-  \nabla \ell(f_{\theta_*}(x_i),y_i)\|^2 = O (\| \theta- \theta_* \| ^2)$ under smoothness assumption. Therefore the variance vanishes when the model approaches a solution: *it corresponds to a gradient estimate without any constant-variance additive noise*. More specifically, in the least-squares setting (in this case: $\ell(f(x),y) = (f(x) - y)^2$ and $f_\theta(x) = \langle \theta, x \rangle$), the existence of a regressor, i.e. $\theta_*$ such that  $\langle \theta_*, x_i \rangle = y_i$, is equivalent to the fact that there is no additive noise in the gradient estimate. Hence, in the finite sum setting, overparameterized models serve as a motive for the no-additive-noise setting in stochastic optimization (see, e.g., *The Power of Interpolation: Understanding the Effectiveness of SGD in Modern Over-parametrized Learning*, Ma et al, ICML, 2018).
> > >
> > > In that sense, we are not trying to motivate *a posteriori* our non-additive noise framework thanks to overparameterization, but on the contrary we were interested *a priori* by no-additive noise because this is the natural framework induced by SGD in modern overparameterized machine learning.
> > >
> > > ---
> > >
> > > **Question:** *Note that overparameterization permits an alternative model assumption; why this could motivate a new data distribution assumption?*
> > >
> > > **Answer:** Thank you for this question that we will need to clarify in the revised version. Indeed, there are two different settings:
> > > - (i)  We can assume that we fit perfectly a finite amount, $n$, of data, i.e.  $f_\theta(x_i) = y_i$ assuming that our model is rich enough to interpolate (in the linear case this simply amounts to have an overparameterized model $d > n$).
> > > - (ii)  We can also assume that the outputs are generated by a deterministic function $f_*$, i.e. such that $f_*(x) = y$ *for all* $x,y$, almost surely. This is a distribution assumption. Note that if, furthermore, we assume that $f_*$ belongs to our model class, this amounts to the existence of $\theta_*$ such that $f_{\theta_*}(x) = y$ for *all* $x,y$, almost surely.
> > >
> > > Your main concern seems to be that we motivate (ii) that corresponds to an assumption on the distribution, by (i) that is simply caused by overparameterization. This is a legitimate concern and you are correct that overparameterization should neither motivate nor be confused with a new data distribution assumption. Though, let us put emphasis on the fact that our framework includes both the finite-sum setting (i) (as briefly outlined in the rebuttal) and the statistical noiseless setting (ii). As the emphasis is put on (ii) in the article let us be more precise for the (i) case:
> > > In this case, the training loss can be written as a finite-sum as follows $f( \theta ) = \frac{1}{n} \sum_i \left( \langle x_i ,\theta - \theta_* \rangle \right)^2 = E_{x \sim D}  \left(  \langle x,\theta - \theta_* \rangle \right)^2$ where $D$ is a uniform distribution over $ \{ x_1 \cdots x_n \} $. Note that with this distribution $D$ over $ \{ x_1 \cdots x_n \} $, our noiseless assumption in the paper holds. Consequently our result on the final iterate convergence holds for the training loss, e.g. according to Theorem 1, the training loss has a final iterate convergence rate of $O( \frac{\ln{T}}{T} )$.
> > >
> > > Therefore, when applied to the finite-data-set case, our assumption should not be read as a data assumption (as explained in the rebuttal, the underlying linear model can just a well be noisy) but instead as a model assumption (there exists an interpolator which fits the *training data*).
> > >
> > > ---
> > >
> > > **Question:** *Perhaps the authors should seek for a different motivation for the studied no-addictive-noise setting.*
> > >
> > > **Answer:** As explained above, our general setting is broader and encompasses:
> > > - *the statistical noiseless setting* where we provide estimation rates which can be, e.g., applied in non-parametric regression (of independent interests, see, eg, Berthier et. al., NeurIPS 2020)
> > > - *the finite sum optimization setting* where we describe the convergence rate of SGD last iterate toward a minimizer of the training loss.
> > >
> > > Both points are valid and important motivations for our work and will be made clearer in the revised version.
> > >
> > > ## With additive noise, constant-stepsizes SGD *is not* converging to the solution
> > >
> > > **Question:** *We all agree that the presented techniques for constant-stepsize SGD cannot deal with additive noise / variance error (in the sense of obtaining a decreasing excess risk).*
> > >
> > > **Answer:** We disagree on this point: our analysis technique *does handle additive noise* however the final iterate of SGD with constant-stepsize *does not converge* as it suffers an asymptotic variance [*Bridging the gap between constant step size stochastic gradient descent and markov chains*, Dieuleveut et al., Annal of Stats, 2020.]
> > >
> > >
> > > ## Relaxed assumptions on the norm of $\theta_*$.
> > >
> > > We hope that the reviewer’s concerns about handling various norms (which was one of the main points raised in the review) has been answered.
> > >
> > > We thank the reviewer for the thorough review, and we are available for any further questions.

---

### Author Response · Authors · 2021-08-31
**We will be happy to discuss any further questions**

Dear Reviewers and Area Chair,

We would like to thank you again for the feedback given in the initial reviews and during the discussion. We hope that we have addressed your main concerns. The discussion phase is ending soon and we remain available should you have any further questions about our work.

Best regards,

 The authors

---

### Decision · Program_Chairs · 2021-09-27

**Decision:**

Accept (Poster)

**Comment:**

This paper proves the risk bound of the last iterate for constant step size SGD in the interpolation regime. The main concern from the reviewers is that the linear regression model is assumed to be noiseless, which makes the results less interesting. After the author response and reviewer discussion, the paper gathers enough support from the reviewers. Thus, I recommend acceptance.